https://doi.org/10.1038/s42003-020-0764-0　　**OPEN**

# Concurrent human TMS-EEG-fMRI enables monitoring of oscillatory brain state-dependent gating of cortico-subcortical network activity

Judith C. Peters [1,2,3,5]*, Joel Reithler [1,2,3,5], Tom A. de Graaf[1,2], Teresa Schuhmann [1,2], Rainer Goebel[1,2,3] & Alexander T. Sack [1,2,4]

Despite growing interest, the causal mechanisms underlying human neural network dynamics remain elusive. Transcranial Magnetic Stimulation (TMS) allows to noninvasively probe neural excitability, while concurrent fMRI can log the induced activity propagation through connected network nodes. However, this approach ignores ongoing oscillatory fluctuations which strongly affect network excitability and concomitant behavior. Here, we show that concurrent TMS-EEG-fMRI enables precise and direct monitoring of causal dependencies between oscillatory states and signal propagation throughout cortico-subcortical networks. To demonstrate the utility of this multimodal triad, we assessed how pre-TMS EEG power fluctuations influenced motor network activations induced by subthreshold TMS to right dorsal premotor cortex. In participants with adequate motor network reactivity, strong pre-TMS alpha power reduced TMS-evoked hemodynamic activations throughout the bilateral cortico-subcortical motor system (including striatum and thalamus), suggesting shunted network connectivity. Concurrent TMS-EEG-fMRI opens an exciting noninvasive avenue of subject-tailored network research into dynamic cognitive circuits and their dysfunction.

[1] Faculty of Psychology and Neuroscience, Department of Cognitive Neuroscience, Maastricht University, P.O. Box 616, 6200 MD, Maastricht, The Netherlands. [2] Maastricht Brain Imaging Center (M-BIC), Maastricht University, P.O. Box 616, 6200 MD, Maastricht, The Netherlands. [3] Department of Vision, Netherlands Institute for Neuroscience, an institute of the Royal Netherlands Academy of Arts and Sciences (KNAW), Meibergdreef 47, 1105 BA, Amsterdam, The Netherlands. [4] Department of Psychiatry and Neuropsychology, School for Mental Health and Neuroscience (MHeNs), Brain+Nerve Centre, Maastricht University Medical Centre+(MUMC+), Maastricht, The Netherlands. [5] These author contributed equally: Judith C. Peters, Joel Reithler. *email: J.Peters@maastrichtuniversity.nl

The principles governing interactions within and between functional neural networks have taken center stage in modern neuroscience, yet many of their fundamental characteristics remain unknown. Likewise, the increasing interest in clinical applications of non-invasive brain stimulation bolster the need for patient-tailored solutions to optimize treatment efficacy; yet specific regimes at the single-subject level are difficult to set up, due to the intricacies of the interactions between the externally applied stimulation and ongoing brain dynamics. There are few available methods that can address such issues on the human systems level, as they require charting network responsivity while simultaneously monitoring variations in brain states. To noninvasively infer causal relations between brain states and large-scale network dynamics, new methods are needed. We previously introduced the simultaneous combination of Transcranial Magnetic Stimulation (TMS), electroencephalography (EEG), and functional magnetic resonance imaging (fMRI) as a potential solution, demonstrating its safety and technical feasibility[1]. Here, we present how this unique multimodal triad of brain imaging and neuromodulation techniques can noninvasively reveal causal dependencies between oscillatory mechanisms and functional connectomics at the single-subject level.

Orchestrated neuronal firing results in rhythmic fluctuations in the excitability of neuronal ensembles[2]. Such oscillatory behaviors impose variations in brain states with functional implications for interareal information transfer, neuroplasticity, and cognitive functions[2–5]. These influences of brain state can be systematically examined by applying TMS during different states, while simultaneously tracking network reactivity[6]. Since intensity, cortical location, and timing of the TMS pulse are under direct, precise experimental control, TMS serves as an excellent systems' probe to evaluate gating of external input under various state conditions. In a multimodal setup, brain state-dependent influences on the propagation of TMS-induced local action potentials via axonal projections can be inferred, providing key insights on effective connectivity[7–9]. For example, a TMS-EEG study by Massimini and colleagues[8] showed that effective connectivity breaks down during non-rapid eye movement (NREM) sleep, as TMS perturbations did not show the spatially widespread and temporally complex cortical propagation patterns typically observed during wakefulness. The present study pursued a similar approach in order to examine TMS-evoked network responses during the meandering sub-states of wakefulness[10], indexed by the well-studied fluctuations in alpha power[11].

A range of neuroimaging studies revealed that increases in alpha power decrease neural activity[12], cortical regional blood flow[13], and fMRI BOLD (Blood-oxygenation-level-dependent) responses[13,14]. These findings suggest that alpha oscillations predominantly serve a suppressive function[12], which appears to be instrumental: alpha power levels predict cortical excitability thresholds[15,16] and task performance[17–20]. According to the "Gating-By-Inhibition hypothesis"[21], alpha oscillations offer a neural routing mechanism that gates incoming information to relevant areas by actively inhibiting irrelevant regions, thereby shunting spurious activity. Therefore, moment-to-moment alpha power (measured with EEG) seems a prime candidate to index fluctuations in regional responsivity, potentially gating the propagation of external impulses (i.e. TMS) throughout a brain network (measured with fMRI). However, causal empirical evidence supporting the influence of oscillatory states on large-scale network dynamics remains scarce.

In the present study, we made use of a concurrent human TMS-EEG-fMRI setup enabling the direct and noninvasive probing of such state-dependent TMS propagation within brain-wide functional systems. This new experimental approach synergistically combines the methodological strengths of its techniques. First, the focal and precisely timed application of TMS allows the controlled perturbation of one network node, functioning as an entry point into the targeted brain-wide functional network. Here, we stimulated the right dorsal premotor cortex (rPMd), a pivotal area involved in high-level motor control[22,23]. Second, EEG has the temporal resolution to capture fast state fluctuations, reflected in electrical signals reaching the scalp. In the current study, EEG was used to track spontaneous fluctuations in parietal alpha-band power prior to the application of TMS. Third, fMRI provides the means to monitor TMS-evoked propagation patterns throughout extended neural networks at high spatial resolution. Including fMRI (besides EEG) is particularly relevant when investigating functional systems with cortico–subcortical substrates, given that surface reflections of subcortical activity in the EEG are absent, or too weak, mixed, and distributed for unambiguous source reconstruction[24,25]. Here, we used fMRI to assess state-dependent modulations in brain-wide network excitability of the cortico–subcortical motor system. In sum, we assessed how pre-TMS EEG alpha power fluctuations influenced activation patterns in the cortico–subcortical motor system, which were induced by subthreshold triple-pulse TMS to the right dorsal premotor cortex.

Results showed that strong pre-TMS alpha power, unlike beta power, shunted network activity in cortico–subcortical motor circuits: both local and remote BOLD responses to TMS perturbation were inversely related to alpha power at the time of the TMS pulse. At the single-subject level, trial-by-trial analyses revealed this state-dependent gating of signal propagation in half of our participant sample. This effect was observed regardless of the functional ROI definition of the motor network (i.e., based on self-generated movements or externally applied TMS). Moreover, this effect was much more pronounced for: (1) TMS-evoked BOLD responses than spontaneous BOLD fluctuations and (2) brain areas inside, rather than outside, the motor system. Overall, these findings show that by assessing the initial state of the motor system (with EEG) when perturbing a key node (with TMS) and monitoring system-wide network dynamics (with fMRI), causal inferences on human brain network functioning can be derived beyond the scope of what various bimodal combinations of noninvasive brain stimulation and imaging techniques can offer. As such, our findings provide a proof-of-principle demonstration of the practical usability of the presented trimodal approach, leading to meaningful observations at the single-subject level. This is crucial, in light of relevant experimental and clinical applications as discussed below.

## Results

**TMS of PMd causes cortico-subcortical motor network activity.** Our first aim was to assess the extent of local and remote motor network reactivity, indexed by fMRI BOLD responses, to transcranial stimulation of the dorsal premotor cortex (PMd), a crucial area for high-level motor control[22,23]. To identify participant-specific right PMd, participants performed a finger tapping task using their left index finger in an initial fMRI localizer session (see Methods for details). This task led to clear activations in the right PMd (Fig. 1a) which could be reliably identified in all participants (Supplementary Fig. 1).

Participants subsequently returned for the combined TMS-EEG-fMRI session (see Methods for used equipment and detailed procedures regarding neuronavigated TMS coil positioning, TMS stimulation settings, and EEG/fMRI acquisition parameters). Here, triple-pulse TMS was repeatedly applied at jittered intervals throughout the scanning session with fixed, relatively low

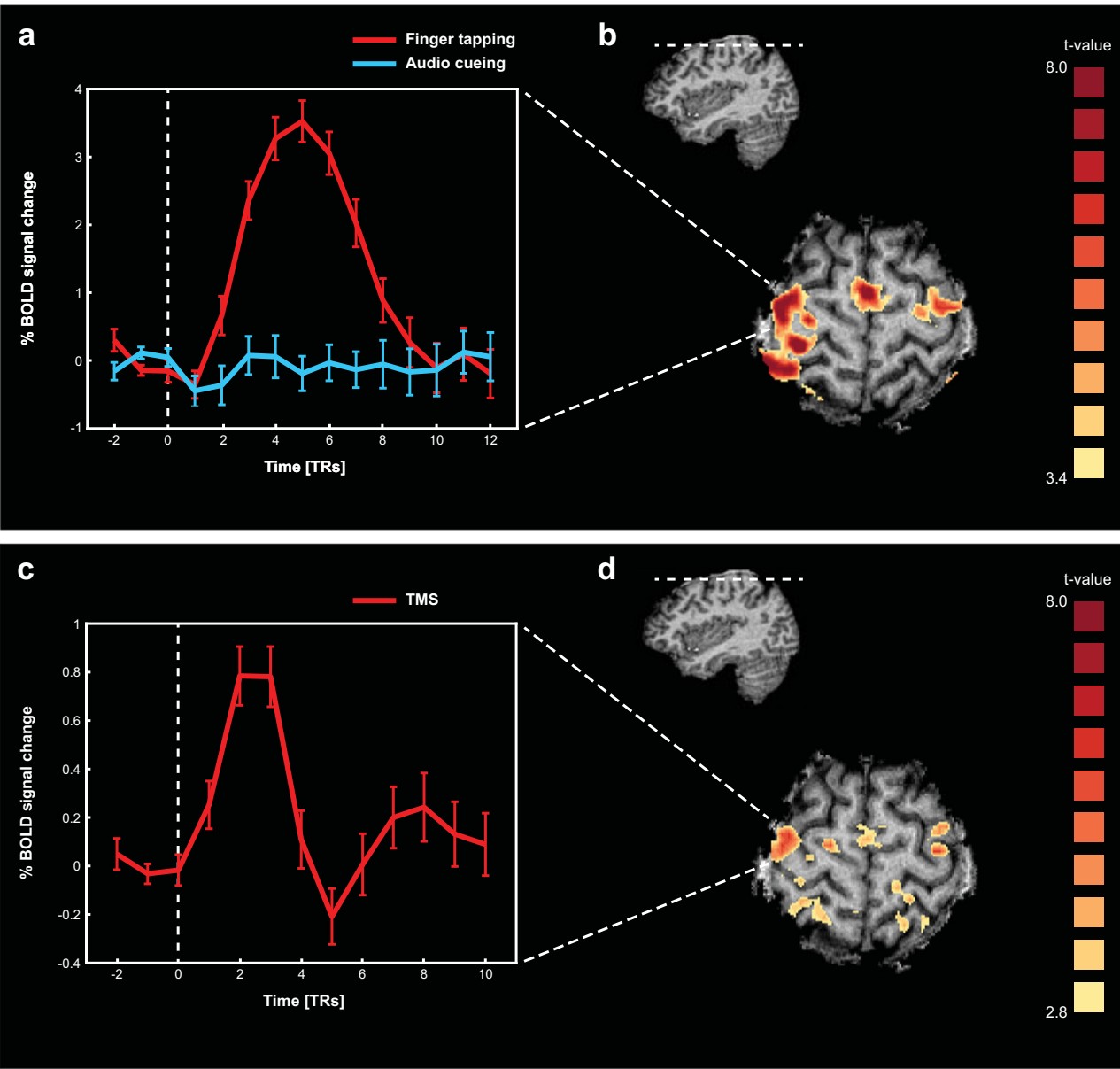

**Fig. 1 Motor network localizer (top) and TMS-evoked responses (bottom). a** Activation of the motor network as observed during the finger tapping task in the first fMRI session, when applying a conjunction contrast ([finger tapping > baseline] & [finger tapping > auditory cueing only]) at FDR(q) < 0.005 (participant 2). Inset shows average response (aligned to movement onset) in the right PMd, which was subsequently chosen as the TMS target site. Unit of x-axis is data points (i.e., in TR, 1 point = 1500 ms). Error bars in this and all subsequent figures represent standard error of the mean (SEM). **b** Position of the transversal slice in **a** shown in the sagittal plane. **c** TMS-evoked BOLD activation map when contrasting all TMS-events vs. baseline (fMRI session 2; participant 2) at FDR(q) < 0.05. Inset shows average TMS-evoked response (aligned to TMS onset) in the right PMd (i.e., the TMS target site). Unit of x-axis is data points (i.e., in TR, 1 point = 2250 ms). These results validate that the neuronavigated TMS coil positioning was in line with the intended target site, and that TMS induced reproducible BOLD responses in the stimulated target site and connected network nodes. **d** Position of the transversal slice in **c** shown in the sagittal plane. Note that the overall shape and timing of the TMS-evoked response matches the typical characteristics of BOLD responses elicited under physiological conditions, when considering the difference in sample rate and the difference in event durations for the motor execution blocks and the TMS applications (i.e., 4 s vs. 133 ms).

stimulation intensity (mean 55% max. stimulator output; MSO) to avoid induction of peripheral motor activity (as this would be accompanied by afferent signals blurring the brain responses of interest[26]). TMS evoked both local and remote BOLD responses (Fig. 1c), reflecting propagation of the focally induced neural activity to structurally and functionally connected areas. The BOLD responses evoked by motor activity (Fig. 1a) and TMS (Fig. 1c) showed similarities in their overall shape and time-course, when taking into account differences in sample rate

and the intrinsic differences between voluntary and TMS-evoked motor activity (e.g., brevity of the TMS-events [133 ms]).

To enable EEG-informed fMRI analyses of network excitability throughout the extended motor system, we employed two complementary approaches to functionally define a set of relevant motor network nodes in each participant. Firstly, we identified the core areas in the motor network engaged by self-generated movement. These "Motor Execution" regions-of-interest (ROIs; "ME ROIs" indicated in purple in Fig. 2) were identified based on

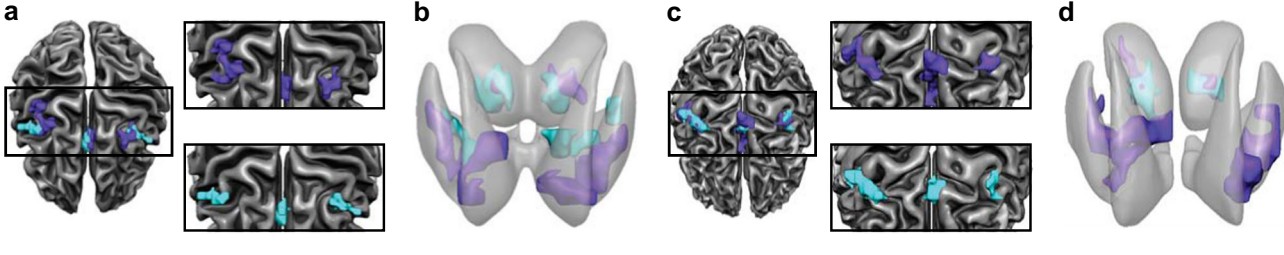

**Fig. 2 Visualization of the employed Regions-of-Interest (ROIs). a** Depiction of the included ROIs in the cortical motor network of participant 1, shown on the corresponding cortical surface reconstruction. In this and subsequent panels, ROIs identified in the independent finger tapping task ('Motor Execution ROIs') are colored purple, whereas ROIs defined by contrasting all TMS-events jointly versus baseline ('TMS-responsive ROIs') are colored cyan. The insets show the isolated Motor Execution ROIs (top) and TMS-responsive ROIs (bottom) to exemplify the extent of overlap across ROI types. **b** Representation of the subcortical ROIs for participant 1, depicted on a semi-transparent rendering of the bilateral striatum and thalamus. **c** Cortical ROIs and cortical surface of participant 2 (formatting identical to **a**). **d** Subcortical ROIs and corresponding anatomy of participant 2.

the independently acquired localizer data from the first fMRI session, and included the right PMd target site, the left PMd, right primary motor cortex (M1), supplementary motor area (SMA), as well as bilateral thalamus and striatum (Supplementary Table 1). Secondly, a further subset of motor regions (with partial overlap with the first set, see Fig. 2) was defined by conjointly contrasting all TMS-events to baseline in the concurrent TMS-EEG-fMRI session ("TMS-responsive ROIs", shown in cyan in Fig. 2). As is evident from the depiction in Fig. 2, the activity evoked by stimulating the right PMd oftentimes nicely aligned with the identified ROIs based on the actual motor task execution.

However, our participants showed strong inter-individual differences in the degree of motor network engagement. Compared to "High Activators" ($n = 2$; 1 female), "Low Activators" ($n = 2$; 1 female) showed relatively low and confined network activations already during Motor Execution (ME), suggesting weaker connectivity (independent of TMS excitation). For example, the mean volume of ME ROIs was more than 3.5 times as small in the Low Activators (mean number of voxels = 31) compared to High Activators (mean number of voxels = 113). Similarly, TMS-evoked network responses differed substantially (being much more confined in low activators), echoing earlier findings on large inter-individual variability in the net effects of TMS manipulations (e.g., only 25% of participants showed all expected effects in[27], also see refs. [28–30] and discussion). It should also be noted that in our sample, TMS was applied at a lower absolute intensity in the Low Activators (mean intensity 45% MSO) compared to High Activators (mean intensity 65% MSO), due to either a low resting motor threshold or participant discomfort. The weaker BOLD responses in Low Activators strongly limited ROI definition: of the 20 potential ROIs (i.e., 10 motor areas defined as ME or TMS ROI), 88% were identified in the High Activators, whereas <50% were present in Low Activators. These results should not be interpreted to indicate fundamental differences between hypothetical subpopulations, but rather support the rationale of pursuing further analyses in a subset of our current sample. For a meaningful evaluation of the state-dependence of TMS-evoked motor network activity, an appreciable motor response overall seems the first requirement, and therefore the EEG-based analyses reported below mainly focused on the identified High Activators subset.

**Strong alpha power impedes TMS-induced signal propagation.** The main question for the current study was: can our methodology reveal on the single subject level how ongoing fluctuations in a participant's brain state influence the propagation of (TMS-evoked) activity throughout a brain network? To index fluctuations in brain state, we used the trial-by-trial variations in

EEG alpha and low-beta power just before the TMS pulse application. Conversely, changes in neural activation due to TMS were assessed by measuring the subsequently induced cortico–subcortical fMRI BOLD response in our predetermined set of ROIs. The link between pre-TMS EEG power estimates and TMS-evoked BOLD amplitude changes was quantified in two ways: first, the correlation between these two measures was assessed across all TMS trials per participant per ROI, and tested against null distributions of random correlations (20000 random-label permutations; non-parametric Wilcoxon sign rank test). Second, we performed "category" analyses comparing the average TMS-evoked BOLD responses when preceded by "weak" versus "strong" EEG power prior to TMS (with binning including 40 trials per power level per participant).

This procedure revealed negative correlations between EEG alpha power and TMS-evoked hemodynamic responses in 77.1% of the ROIs (i.e., 27 of 35 ROIs) of High Activators (see Supplementary Table 1 for details). Thus, stronger parietal alpha power at the time of stimulation resulted in lower TMS-evoked BOLD responses throughout the motor network. This conclusion was supported by the second, more powerful, category analysis. Here, a large majority of ROIs (88.6%; i.e., 31 of 35 ROIs) showed a reduced TMS-evoked BOLD response following a strong compared to weak pre-TMS alpha power state (see Supplementary Table 1 for details). This impediment of the TMS-induced signal propagation was seen in both ipsi- and contralateral regions relative to the TMS target site. Remarkably, both cortical (Fig. 3) and subcortical (Fig. 4) nodes in the motor network were similarly affected.

To evaluate the specificity of the alpha-dependent variations in BOLD amplitude, we additionally performed the same analyses on (1) interspersed null-events (in which no TMS was administered) and (2) BOLD activity in non-motor ROIs (i.e., the angular gyrus, primary auditory cortex, and orbitofrontal cortex; see methods for details). Relatively few of our main motor ROIs showed an influence of parietal alpha power on spontaneous hemodynamic activity as observed during null events (correlation: 7 of 35 ROIs = 20%; category: 13 of 35 ROIs = 37.1%), indicating that the observed alpha modulation of TMS-evoked propagation patterns is not simply a reflection of ongoing covariations between alpha power and spontaneous BOLD fluctuations. Moreover, pre-TMS alpha power did not affect fMRI activity in the non-Motor ROIs to the same extent as in Motor-ROIs: The angular gyrus did not show any alpha-dependent BOLD variations (all $p$'s > 0.05), except for the correlation analyses of TMS-events in participant 2 ($p = 0.023$). In addition, BOLD activity in the primary auditory cortex was either not affected by alpha power (participant 2: $p$'s > 0.05), or

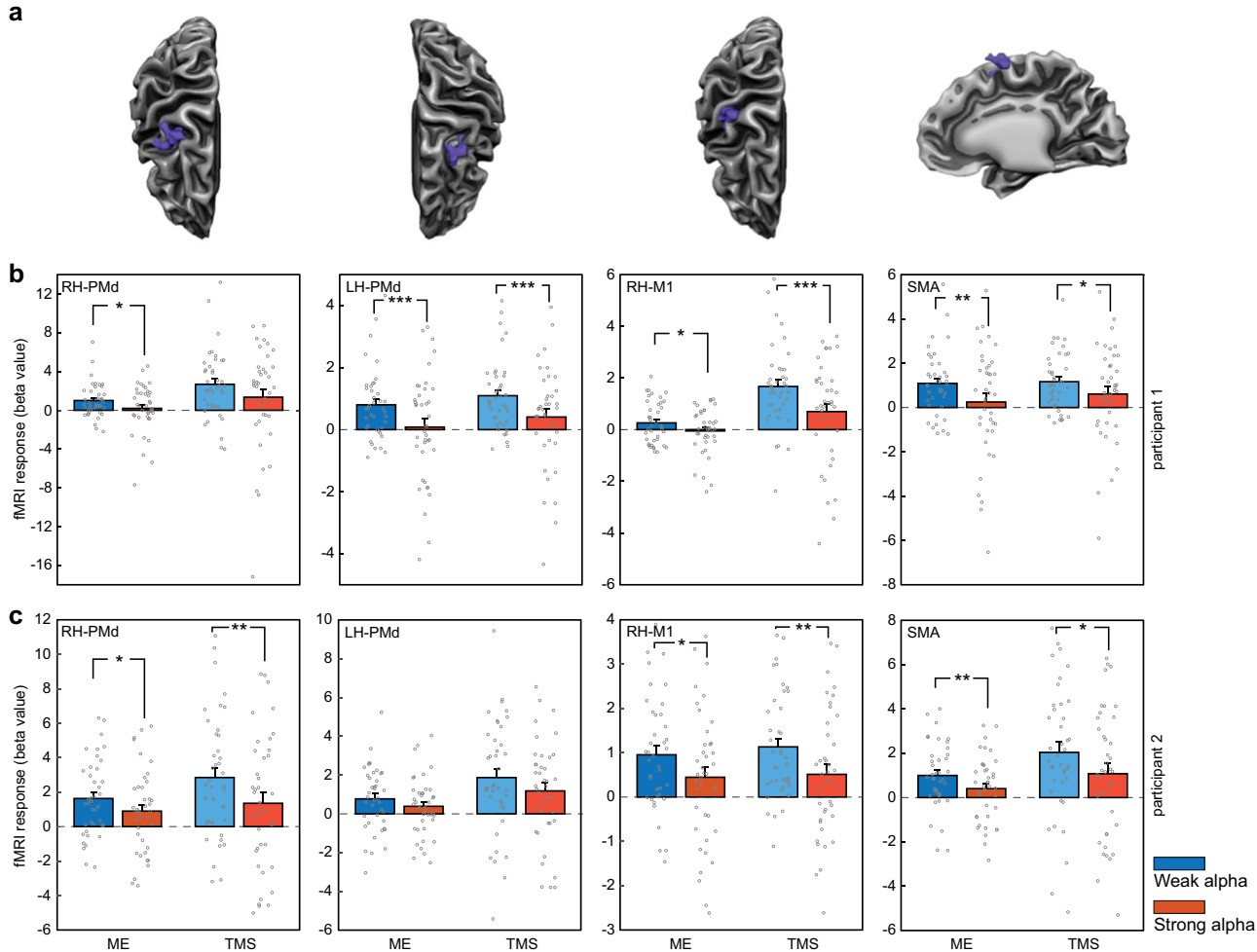

**Fig. 3 TMS-induced BOLD changes in cortical ROIs depend on pre-TMS alpha power. a** The cortex mesh reconstructions show the location of the cortical 'Motor Execution' ROIs (in participant 1) corresponding to the ROI fMRI data in the box plots below. **b** Depicted are the TMS-evoked BOLD responses (indexed by their average beta value) in the specified ROIs when the pre-TMS alpha power was either "weak" (blue bars) or "strong" (red bars; $n = 40$ trials per category). Circles represent individual trials. Data from participant 1. **c** Data from participant 2 (formatting and color-coding are identical to **b**). A clear pattern is visible in which weak alpha power preceding the stimulation event leads to stronger BOLD activations at the TMS target site and connected cortical motor network nodes. Note that the darker and lighter colored bars represent the fMRI activation levels in the "Motor Execution" (ME) and "TMS-responsive" (TMS) ROIs, respectively. Significance levels: $*p < 0.05$; $**p < 0.02$; $***p < 0.01$. PMd dorsal PreMotor cortex, M1 Primary Motor Cortex, SMA Supplementary Motor Area, RH right hemisphere, LH left hemisphere.

showed a similar modulation for Null- and TMS-events (participant 1: null-events category: $p = 0.017$; null-events correlation: $p = 0.010$; TMS-events category: $p = 0.004$; TMS-events correlation: $p > 0.05$). Finally, orbitofrontal activity was not influenced by pre-TMS alpha power ($p$'s $> 0.05$). Yet, it was affected by alpha power prior to null-events (in participant 1 only: Null-events category: $p = 0.042$; Null-events correlation: $p = 0.012$). In sum, we mainly observed an absence of TMS-effects in non-motor ROIs (correlation: 1 of 6 ROIs; category: 1 of 6 ROIs). Moreover, these effects appeared to be rather unspecific, as we observed similar effects for Null-events (correlation: 2 of 6 ROIs; category: 2 of 6 ROIs). These lacking or unspecific effects contrast with the marked and specific influence of pre-TMS alpha power on activity in (the large majority of) Motor ROIs. This suggests that if there is any relation at all between alpha power and BOLD activity in the non-motor ROIs, it is unlikely to be related to the TMS-stimulation. Together with the absence of effects in some motor ROIs (Supplementary Table 1) and the stronger presence of alpha-dependent influences on TMS-evoked than spontaneous BOLD responses in the motor ROIs (null-event analyses), these findings indicate that the identified alpha

power-driven BOLD modulations are rather specific to the signal propagation in the motor network targeted by TMS.

In addition, to further investigate the specificity of this negative relation between pre-TMS alpha power and TMS-evoked activity, we applied the same analyses using pre-TMS beta power fluctuations (estimated at the right PMd site) as independent variable (Supplementary Fig. 3). In contrast to alpha activity, strong pre-TMS beta power did not systematically shunt propagation of TMS excitations throughout the targeted network.

## Discussion

Adaptive brain function requires the integration of segregated brain circuits into transiently reconfigured functional networks. This continuous coupling and uncoupling of functional networks results in ongoing fluctuations in brain states. In turn, these states affect the brain's excitability by sensory input[2]. Here, we present how human TMS-EEG-fMRI measurements can contribute to disentangling the intricate interplay between local oscillations, neural excitability, and brain-wide network dynamics. More specifically, we assessed how trial-by-trial cortical EEG alpha

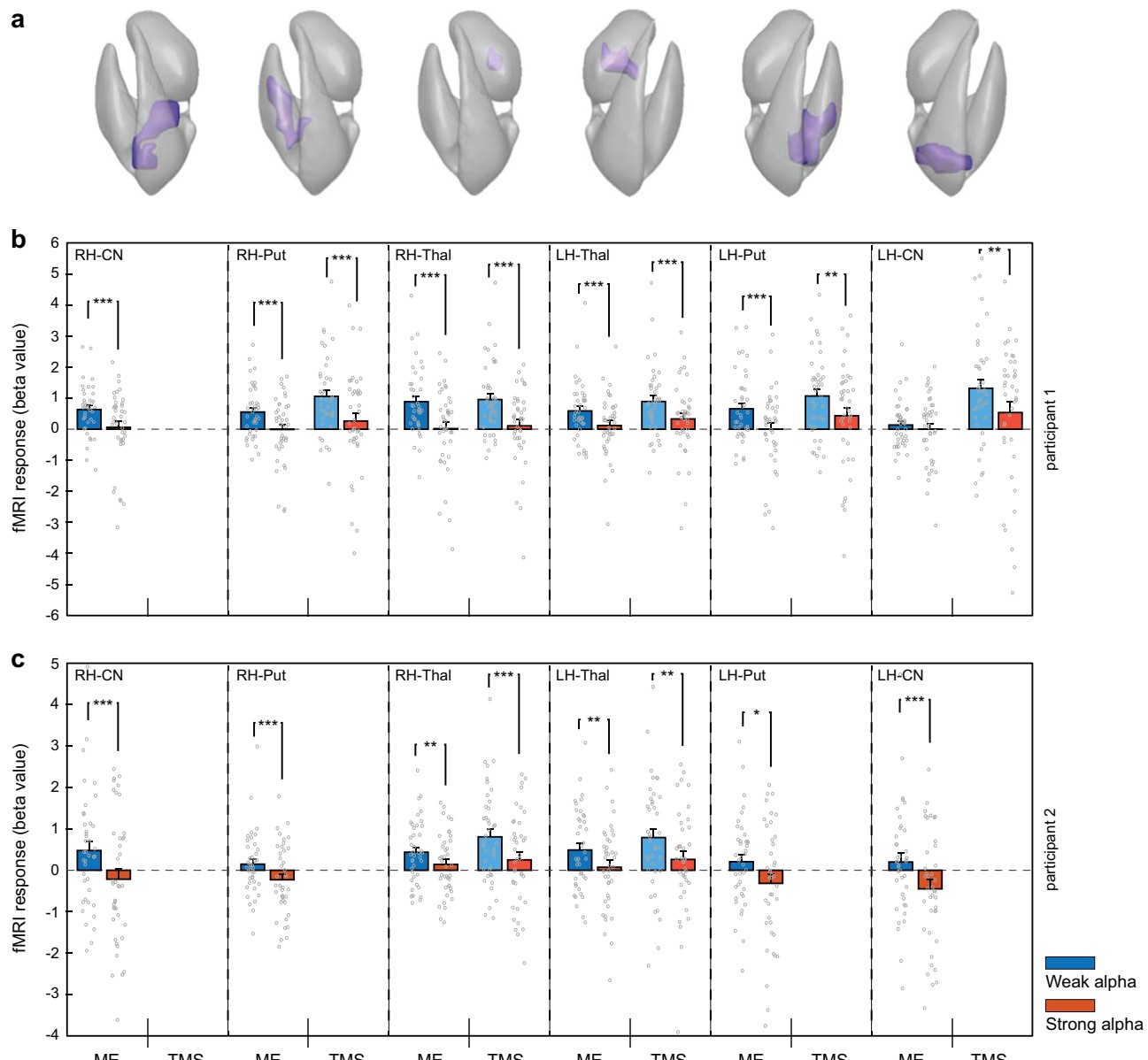

**Fig. 4 Pre-TMS alpha power variations also affect TMS-induced BOLD changes in subcortical motor regions.** The modulatory effect of the pre-TMS brain state (as reflected through variations in parietal alpha power) is also observed in subcortical areas, highlighting that also remote TMS effects are more pronounced when pre-TMS alpha power is relatively weak. Formatting and color-coding are identical to Fig. 3 ($n = 40$ trials per category). **a** Mesh reconstruction of the striatum and thalamus, depicting the location of the subcortical "Motor Execution" ROIs (in participant 1) corresponding to the ROI fMRI data in the box plots below. **b** Data from participant 1. **c** Data from participant 2 (formatting and color-coding are identical to **b**). Missing bars indicate that the ROI could not be determined in the corresponding participant. CN Caudate Nucleus, Put Putamen, Thal Thalamus.

fluctuations modulated TMS-evoked activations of the cortico–subcortical motor system. Subthreshold triple-pulse TMS to the right dorsal premotor cortex activated a bilateral network of cortical and subcortical areas (including thalamus and striatum) that was also recruited by self-generated movement. The subsequent EEG-informed ROI analyses revealed that, in participants with sufficiently strong motor-network BOLD responses overall ("High Activators"), strong pre-TMS alpha power resulted in decreased TMS-induced BOLD activations throughout this network, suggesting shunted network connectivity. This observation is in line with the notion of alpha oscillations acting as an inhibitory mechanism[12,21], directly influencing how strongly widespread network nodes respond to controlled local TMS perturbations. In the same frequency-band as parietal alpha, several studies have investigated the influence of mu-phase in

motor cortex on motor evoked potential (MEP) amplitude with an EEG-TMS setup. Larger MEP amplitudes were found when TMS was administered at the trough compared to when it was administered at the peak of the mu-oscillation[31–33] (but see ref. [34]). Others investigated the role of mu-power on MEP amplitude, suggesting that sensorimotor mu-power may reflect a net facilitation or disinhibition of M1[35]. The intricacies of how peripheral consequences (in terms of MEPs) relate to changes in inter-areal communication at the cortical level, and the role of local mu and global (parietal) alpha therein, remain to be further elucidated.

In stark contrast to alpha activity, strong pre-TMS beta power did not impede the propagation of TMS excitations throughout the motor network. This result supports a divergent influence of beta and alpha oscillations and provides a meaningful frequency-

specificity constraint on the alpha power results. Although outside the scope of this study, a facilitating rather than inhibitory role in signal propagation for beta-band oscillations would match and extend recent insights in the key role of beta activity in coordinating communication in the human motor system[36,37], a role recently shown to be causal[36]. In fact, beta oscillations are the dominant frequency in primary motor cortex at rest[38], and govern cortical control of motor output[39,40]. EEG beta synchronizes with electromyography (EMG) from hand muscles during motor tasks, suggesting a crucial role in sensorimotor integration and connectivity[41,42]. Stimulating motor cortex activity with single-pulse TMS induces beta oscillations[43] and electrical stimulation at beta frequency (but not alpha) modulates motor cortex excitability[36,37], in one case specifically for low-beta frequencies[44], and can change connectivity patterns[45].

Overall, our current results largely align with previous findings, reaffirming that the selected frequency-bands and targeted cortico–subcortical motor network were suitable candidates to provide a proof-of-principle demonstration showcasing one application of this novel multimodal setup: TMS-EEG-fMRI can, in individual participants, reveal trial-by-trial EEG state modulations of network-wide signal propagation with high spatial detail. This interesting finding, that the impact of TMS pulses on cortico–subcortical motor circuits depends on current cortical oscillatory state, can be further investigated and utilized in a range of novel applications.

In recent years, combining non-invasive brain stimulation with fMRI or EEG has become increasingly popular[6,46]. Concurrent TMS-fMRI (and TMS-PET) studies have provided key insights in large-scale effective connectivity, by stimulating one area and monitoring the subsequent activity changes in extended brain networks[47–50]. This method is especially fruitful for imaging activity propagation in cortico–subcortical circuits[51,52], whose sources are difficult to reconstruct from EEG (due to weak, distributed surface reflections of entangled sources and sources with closed-field geometry[24,25]). However, concurrent TMS-fMRI studies cannot capture the fast temporal dynamics of ongoing neural communication within and between interconnected networks. Even during rest, there is continuous coupling and uncoupling of functional networks, resulting in ongoing fluctuations in brain states. Since such fluctuations (e.g., in alpha power) affect network excitability[12,16,21], it is crucial to interpret TMS-induced network activations in a temporal context. Therefore, measuring EEG in addition to fMRI is essential to comprehend how oscillations and brain-wide network dynamics interrelate. Moreover, although concurrent EEG-fMRI studies (without TMS) undoubtedly contributed to our understanding of associations between EEG oscillations and BOLD activity[13,14], causal dependencies between these two modalities remain elusive. TMS constitutes an ideal means to causally and precisely manipulate network activity, as intensity, location and timing of stimulation are under direct experimental control. In sum, only the concurrent combination of TMS, EEG, and fMRI provides a non-invasive method to systematically control (TMS) and visualize system-wide network dynamics (fMRI), while also assessing the initial network state at the onset of stimulation (EEG).

Such monitoring of activity trajectories following network perturbation at varying initial states, could lead to valuable novel insights into (pathological) network behavior, as it provides experimental constraints that can further inform recent efforts to capture and mechanistically interpret such dynamics within formal mathematical frameworks (such as network control theory[53–56]). Futhermore, future concurrent TMS-EEG-fMRI studies on the propagation of TMS-induced activity in functional networks could examine a wide range of related hypotheses[57], through flexible adaptation of the oscillatory band, TMS protocol, or cognitive task of interest, while stimulating network nodes with varying functional relevance.

In addition, TMS is increasingly used for therapeutic applications, e.g., for treating depression[58] or for supporting recovery during neurorehabilitation[59]. It has long been known that while the direct stimulation effects of TMS are limited to superficial cortical brain sites, cortical TMS stimulation often leads to indirect, remote effects in connected subcortical structures as induced activity propagates via axonal projections[60]. In fact, many behavioral effects in fundamental neuroscience research[61], as well as therapeutic effects in the clinical applications of TMS, depend on these indirect effects following the propagation of cortical TMS excitations through such cortico–subcortical networks. The local and remote TMS impact, however, is causally dependent on the state of neural excitability during application. Yet, until now it was impossible to control or modulate the direction and strength of these local and remote, state-dependent network effects of TMS. Simultaneous human TMS-EEG-fMRI holds the potential to overcome this limitation, allowing future EEG-triggered applications to apply TMS at predefined oscillatory brain states that either will or will not lead to specific remote network brain activations by either facilitating or shunting signal propagation from the cortical TMS stimulation site[31]. The current demonstration that this feature can be exploited at the single-subject level will be particularly relevant in clinical contexts where patient-tailored approaches will be required that go beyond average group statistics. It should be noted that the presented modulatory effect of pre-TMS alpha power could only robustly be shown in a subset of our subject sample (in line with earlier work underlining a substantial degree of variability in reported TMS effects). Although this might limit the potential clinical translation to specific patient cohorts, it also provides opportunities to estimate which subpopulation of patients might ultimately benefit most from brain stimulation treatments.

In conclusion, the present study establishes the unique potential of human TMS-EEG-fMRI applications to reveal brain-wide dynamics underlying excitability and signal gating. More specifically, our findings reveal that alpha power fluctuations can causally affect responsivity in the subcortical and cortical motor system. Future extensions of this versatile multimodal approach to other sets of functional networks, task paradigms, stimulation regimes, and participant populations may aid causal inferences on the interplay between oscillatory brain states and subcortical network dynamics in both health and disease.

## Methods

**Participants.** Four healthy, right-handed volunteers (two men; mean age 26.3 ± 2.7 years) participated in the current study. All subjects were members of the academic community of Maastricht University who did not meet any of the MRI or TMS[62] exclusion criteria and who had previously participated in TMS, EEG, and fMRI experiments. Written informed consents were collected in accordance with the Declaration of Helsinki, and all procedures were approved by the Medical Ethical Committee of the Maastricht University Medical Centre.

**Concurrent TMS-EEG-fMRI setup.** While measuring 3T fMRI, EEG, and EMG (to validate absence of physical motor output), we applied triple-pulse TMS to the right dorsal premotor cortex. The participant-specific target site was based on independent fMRI localizer data and coil-placement was guided by TMS-neuronavigation techniques[63] (Fig. 1a, Supplementary Fig. 1). Subsequently, TMS-evoked BOLD activity was analyzed in the extended, functional motor system as a function of the alpha and beta power in the EEG interval immediately prior to TMS application. Detailed descriptions on how the safety and data-quality of concurrent TMS-EEG-fMRI was assessed (in phantom and human measurements), and practical recommendations for safety and quality assurance of adapted setups, are provided in an earlier publication by our group[1].

**MR acquisition.** Each subject participated in two MRI sessions where functional (echo-planar, EPI) and anatomical images were acquired using a standard birdcage

coil (USA Instruments, Aurora, USA) on a 3T head-only Magnetom Allegra MR scanner (Siemens, Erlangen, Germany).

The first "Motor Network Localizer" session entailed a standard motor-activation localizer task to identify the motor network for subject-specific TMS target site selection. In this task, participants tapped their left index-finger paced by 500 Hz sine tones (presented every 250 ms for 150 ms) during "Motor Execution" intervals (4 s each; 26 blocks in total). These were interleaved with 12, 15 or 18 s rest periods. A second run was acquired to serve as control for auditory stimulation. In this run, the same auditory pacing tones were presented while subjects were instructed not to perform the corresponding finger taps. The order of these two runs (of 323 volumes each) was counterbalanced across subjects. In the final run (comprising 560 volumes), participants rested with their eyes open (first half) or closed (second half; not included in further analyses). Next to these functional runs (EPI; 24 transversal slices; TR/TE = 1500/30 ms; FA = 71°; bandwidth = 2112 Hz/Px; 3.5 mm isotropic nominal resolution), a T1-weighted high-resolution anatomical dataset (magnetization-prepared rapid acquisition gradient echo sequence [MPRAGE]; 192 sagittal slices; FoV 256 × 256 mm; TR/TE = 2250/2.6 ms; FA = 9°; 1.0 mm isotropic nominal resolution) was acquired.

In the second "TMS-EEG-fMRI" session, we collected 4 ($n = 2$) or 5 ($n = 2$) resting-state runs (EPI; 21 transversal slices; TR/TE = 2250/30 ms; gap = 750 ms; FA = 78°; bandwidth = 2112 Hz/Px; 3.5 isotropic nominal resolution; 313 volumes per run) and a lower-resolution T1-weighted anatomical dataset (MPRAGE; 192 sagittal slices; FoV 128 × 128 mm; TR/TE = 2250/2.6 ms; FA = 9°; 2.0 mm isotropic nominal resolution). In each run, TMS was applied in gap intervals during functional image acquisition, whereas EEG and EMG were recorded throughout the run.

**EEG and EMG acquisition**. EEG data were recorded via two MR-compatible "BrainAmp MR plus" amplifiers (16-bit A/D conversion; 0.5 µV resolution; ± 16.384 mV operating range; 5000 Hz sampling rate) powered by MR-compatible, rechargeable PowerPacks (Brain Products, Munich, Germany). Both amplifiers were placed next to the scanner's rear end in order to use connection cables with a minimal length. To prevent noise (e.g., cable motion), these flat-wire cables were straightened, aligned with the main z-axis of the scanner bore, and subsequently fixed with sandbags. Data were collected using a 64-electrode TMS- and MR-compatible EEG cap (EasyCap, Herrsching, Germany). The cap contained 62 scalp electrodes (equidistantly spaced and in accordance with the extended International 10/20 system), with AFz functioning as ground and Cz as online reference. The additional two electrodes were placed under the right eye and at the lower back to record EOG and ECG, respectively. Electrodes were sintered Ag/AgCl ring electrodes equipped with current limiting 5 kΩ (scalp) or 15 kΩ (ECG and EOG electrode) resistors with low power dissipation.

The EMG was acquired from two sintered Ag/AgCl ring electrodes placed at the left first dorsal interosseus, and a third at the left processus styloideus radius as off-line reference. Their signal was amplified by an additional "BrainAmp ExG" amplifier (Brain Products, Munich, Germany) and integrated with the EEG recordings using BrainVision Recorder software. This software was running on a PC outside of the MRI room to which the amplified signals were transmitted through a waveguide using fiber-optic cables. EMG was monitored during (TMS-EEG-fMRI) resting state runs to off-line detect any Motor Evoked Potentials (MEPs) that might have occurred from TMS stimulation. Note that with the current stimulation intensity and TMS coil position, we did not expect any MEPs[64] (also see next section). EEG and EMG acquisition (BrainVision Recorder) were synchronized to the MRI scanner's internal clock driving the gradient switching system using a SyncBox Interface (Brain Products, Munich, Germany). Electrodes were filled with conductive gel (Abralyte 2000) and electrode impedance was kept below 30 kΩ for all electrodes.

**TMS setup and TMS neuronavigation**. The TMS setup consisted of an MR-compatible figure-of-eight TMS coil (MagVenture "MRI-B88 coil" without any custom modifications), connected via a 6.2 m cable to an RF filter box located in the cabinet room and directly attached to a waveguide running through the MRI room's wall. The filter box's other end was connected to a MagPro X100 stimulator (MagVenture, Farum, Denmark), equipped with a supplemental module to ensure no artificial MR signal fluctuations were induced due to leakage currents[65]. In the scanner room, we determined for each participant the appropriate TMS stimulation intensity following a procedure introduced by previous TMS studies targeting dorsal premotor cortex[50,66] (except that we established the target site using subject-specific neuronavigation rather than based on scalp distance estimates). First, we assessed the resting motor threshold (rMT), as measure of functional impact of TMS at the level of the motor cortex[64]. To this end, we stimulated the right primary motor cortex adjusting intensity until a visible movement of the resting contralateral index finger or thumb was elicited in 50% of the trials (see ref. [64] for procedural details). Subsequently, we placed the TMS coil above the dorsal Pre-Motor area, which was functionally defined for each individual subject based on the previously acquired "Motor Network Localizer" data (Supplementary Fig. 1). The EEG cap location on which we centered the TMS coil was established using standard (offline) neuronavigation procedures for optimal stimulation results[63]. The TMS coil was finally tightly fixed with cushions to ensure its position while scanning. TMS pulse triplets ("TMS-events") were delivered at 15 Hz every 13.5 s

(with a temporal jitter in the range between 11.25 and 15.75 s) on average at 55% of maximum stimulator output (individually titrated based on the determined rMT). To ensure that our potent triple-pulse TMS over PMd did not elicit muscle responses, TMS intensity was fixed 5% (in terms of MSO) below rMT. For one low activator participant, intensity was lowered slightly more to accommodate participant discomfort, whereas for a second low activator participant intensity was set at rMT to accommodate our lower bound on machine output intensity (45% MSO). During the fMRI measurement, an experimenter was present in the MR scanner room throughout the experiment to monitor potential twitches in the left hand. No muscle twitches were observed during any of the measurements (see EMG analyses below). For analyses purposes, we also collected "Null-events", which were presented in interleaved fashion with TMS-events but during which no TMS stimulation was delivered. Null-events were matched to TMS-events in terms of their number of occurrences and followed a similar temporal distribution. TMS administration was triggered by TTL-pulses sent by a stimulus presentation software package (Presentation v14, Neurobehavioral Systems, Albany, CA, USA), time-locked to the onsets of the MR volume acquisitions (i.e., taking place 500–600 ms after EPI acquisition offset).

## Data analyses

**MRI and fMRI analyses**. Preprocessing of all fMRI data included slice scan time correction, linear trend removal, high-pass filtering, and three-dimensional motion correction as implemented in the BrainVoyager software package[67] (Brain Innovation, Maastricht, the Netherlands). The first two volumes of each run were discarded to remove T1 saturation effects. Functional data (resampled to $3 \times 3 \times 3$ mm³ voxel resolution) were not spatially smoothed, but a temporal high-pass filter was applied to remove frequencies lower than two cycles per time course. All functional data were aligned to the high-resolution anatomical dataset acquired in the first imaging session. In addition, mesh cortex reconstructions were created to guide the neuronavigation procedure prior to the second scanning session.

Whole-brain single trial responses were estimated across runs by fitting a General Linear Model (GLM) to each voxel's individual time course. The design matrix included 2-gamma HRF-convolved stick predictors to model TMS-events, six head motion confound predictors (i.e., estimates of head rotation and translation), and one constant predictor to account for baseline shifts across runs. A similar GLM was used to estimate responses to Null-events, with TMS-events aggregated in a single predictor. The data were z-normalized and corrected for serial correlations using a second-order autoregressive model.

**ROI definitions**. "Motor Execution ROIs" (ME ROIs): for each participant, a whole-brain GLM was calculated on the motor network localizer data. Movement-induced activation of the motor system was assessed by a conjunction contrast ([motor-execution > rest] and [motor execution > auditory tones only]), based on boxcar predictors convolved with a 2-gamma hemodynamic response function (HRF). The resulting statistical map was subsequently used to define "Motor Execution" ROIs, i.e., clusters in the motor system that showed enhanced activity during motor execution intervals (q(FDR) < 0.05; cluster volume ≥ 10 voxels). These ROIs played two key roles in our study: first, the right dorsal PreMotor (PMd) cortex ROI served as target site for TMS neuronavigation. By visualizing the TMS stimulation focus with respect to the subject-specific and functionally defined PMd region superimposed on the individual's cortex reconstruction (Supplementary Fig. 1), we aimed to position the TMS coil optimally to directly target the motor execution network. Second, by assessing the effect of pre-TMS EEG power variations on TMS-evoked BOLD responses in these ROIs, we were able to establish state-dependent TMS propagation in the independently, functionally defined, motor execution system.

"TMS-responsive ROIs" (TMS ROIs): for each participant, TMS-evoked hemodynamic responses in the 'TMS-EEG-fMRI' resting-state data were modeled with 2-gamma HRF-convolved

stick predictors and subjected to a whole-brain GLM. A set of "TMS-responsive ROIs" was subsequently defined for each subject based on TMS-evoked activations across all TMS-events jointly (q(FDR) < 0.05; cluster volume ≥ 10 voxels) in anatomical structures broadly associated to the motor system.

Together, these two sets of ROIs congruently indicated 10 areas responsive to motor execution demands and TMS stimulation of the right PMd (see Supplementary Table 1): right (r) and left (l) dorsal PreMotor cortex (rPMd, lPMd), supplementary motor area (SMA), bilateral caudate nucleus (rCN, lCN), bilateral putamen (rPut, lPut), bilateral Thalamus (rThal, lThal), and right primary motor cortex (rM1).

"Non-Motor ROIs": to establish the specificity of TMS-evoked activity and its relation to pre-TMS power, we defined an additional set of control ROIs. To this end, we transformed the voxel coordinates of Brodmann area 11, 39, and 41 from Talairach-space to the individual anatomical space of each participant. Brodmann area 11, 39, and 41 are part of the bilateral orbitofrontal cortex (OFC), angular gyrus, and primary auditory cortex (PAC), respectively. These ROIs were selected to obtain a heterogeneous sample of control ROIs, both in terms of their varying relations with our research objective as well as their spatial dispersion across the brain (confined by a joint spatial coverage of functional data in all participants): as anterior, medial brain sample, we selected OFC, which is involved in multiple functions including decision-making, response inhibition, and contingency learning operations[68]. The engagement of OFC in these functions implies an indirect, reciprocal relation to the motor system at the highest level of action plan formation and response adaption. In contrast, the Angular Gyrus is located in posterior, lateral cortex and has no obvious relation to the motor system. Finally, PAC in the temporal cortex, was selected as a control for a side effect of TMS stimulation: PAC is not part of the motor system and therefore unlikely to show reactivity to the TMS stimulation itself. However, the auditory "click" sounds of the TMS system during pulse delivery were expected to elicit BOLD activity in PAC[47].

**EEG and EMG analyses.** EMG and EEG data were processed using Vision Analyzer 2.1 (Brain Products, Munich, Germany). The electrophysiological time windows of interest around TMS pulse application were orthogonal to fMRI acquisition intervals (i.e., during a 750 ms acquisition gap), and therefore contained no MRI-gradient induced artifacts. EMG dorsal interosseus bipolar channels were locally referenced and segmented into 160–260 ms epochs after each application of the first TMS pulse (the entire triple pulse TMS event lasted for 133 ms hampering signal quality) or an equal number of matched null events (i.e., epochs with similar timing with respect to MRI acquisition and dispersion across the run as TMS-events, but which were not preceded by pulses). MEPs were automatically detected (>±5 SD of 0–10 ms baseline; search intervals: onset = 5–40 ms; peak = 15–50 ms) in these epochs, using the "EMG Onset Search v1.3" algorithm implemented in Vision Analyzer. The number of intervals that exceeded this threshold did not differ between TMS-events (mean 7.75; s.e. 1.2) and null-events (11.50; s.e. 6.4; Wilcoxon sign-rank p = 0.875).

EEG data were downsampled (500 Hz), re-referenced to an average electrode (i.e., average of F1, F2, FC1, FC2, F3, F4, FC3, FC4, Fz, FCz, C1, C2, C3, C4, CP1, CP2, CP3, CP4, CPz, P1, P2, Pz), epoched (−500 to −5 ms before application of the first TMS pulse), baseline corrected (−450 to −50 ms), detrended, and filtered with a Butterworth Zero Phase band-pass (0.5–40 Hz; 24 dB/oct slope) and Notch (50 Hz) filter. Finally, epochs of electrodes containing artefacts (i.e., automatically detected intervals with steep gradients [20.00 μV/ms or a difference of 200 μV per 50 ms interval] or exceeding ±80 μV) were excluded

from the subsequent pre-TMS power analyses. To perform power analyses, we exported the preprocessed data of the right hemispheric, frontal FC4 and parietal P2 electrode to Matlab 2018a. This hypothesis-driven[36,69] selection of electrodes was used to assess beta power variations at the TMS-target site (i.e., right PMd) and alpha power variations at the parietal site (a scalp location close to the well-established major parietal alpha source[69]), respectively. For each of the two sites, single-trial 8–18 Hz power spectra were obtained by detrending, and Fourier transforming (Hanning-tapers; moving time window of three cycles per frequency; 2 Hz frequency resolution) each pre-TMS epoch using the FieldTrip toolbox[70]. The relatively low spectral resolution hampered the extraction of individual alpha and beta frequency peaks. Therefore, the frequency with the maximum power in the alpha (8–12 Hz) and low-beta (14–18 Hz) range of the mean power spectrum were used (see Supplementary Fig. 2). Trial-by-trial fluctuations at this alpha peak (10 Hz) and beta peak (14 Hz) frequency served as indices of alpha power and beta power, respectively, in the subsequent EEG-fMRI analyses.

Recent findings demonstrate that alpha power can show slow drifts over the course of an experiment, impacting behavioral measures[71,72]. We tested the correlation between alpha power on null-events and trial order but found no evidence of a drift in alpha power estimates across trials (all p's > 0.05). One reason for this discrepancy could be that previous studies were performed within a behavioral task context involving many trial repetitions, possibly leading to systematic fluctuations in terms of fatigue, overall vigilance or motivation. In contrast, in the current experiment participants were not engaged in any demanding behavioral task and consecutive runs were relatively short, which might have led to a more homogeneous state in terms of mental engagement. Nevertheless, this is an important factor which should be considered when moving forward towards task-based experiments involving attention-demanding behavioral tasks.

**EEG-informed fMRI analyses.** To examine how TMS signal propagation was affected by alpha and beta power fluctuations prior to TMS stimulation, we performed two types of ROI analyses: First, we computed for each participant the correlation of pre-TMS EEG power and TMS-evoked BOLD activity across trials. Then, we assessed the significance (p < 0.05) by contrasting the obtained Pearson's rho values against null distributions of those values based on 20000 permutations of the TMS-evoked activity estimates. Second, we clustered trials according to their pre-TMS power in a "weak-power" and "strong-power" category (bin of 40 trials per category) and compared the TMS-evoked hemodynamic activity between these two categories. Significance was estimated by applying the same permutation procedure as used in the correlation analyses (20000 permutations; p < 0.05). Note that to further characterize the relation between pre-TMS power and TMS-evoked activity, these analyses were not only applied to TMS-events, but also to null-events. Moreover, EEG-driven effects on BOLD responses to TMS-events and null-events were examined in both Motor-ROIs as well as an additional set of Non-Motor (control) ROIs.

**Statistics and reproducibility.** To recapitulate, fMRI data were analyzed voxel-by-voxel using whole-brain GLMs based on boxcar (localizer) or stick (TMS- and null-events) predictors convolved with a 2-gamma HRF. P-values of all voxels were adjusted for multiple comparisons using the False Discovery Rate (FDR), applying a proportion of false discoveries of q = 0.05 (corrected p values are indicated as "FDR(q)")[73]. For EMG data, an automatic MEP detection was performed for the preprocessed event intervals. We statistically compared MEP occurrences of

TMS- and null-events using the non-parametric Wilcoxon sign-rank test. Furthermore, potential drifts in EEG alpha power, over the course of the experiment, were assessed by computing the Pearson's linear correlation coefficient (rho) between alpha power and trial order and transforming rho to the corresponding $p$-value using a Student's $t$ distribution.

For the EEG-informed fMRI analyses, two types of analyses were performed. First, Pearson's correlation between pre-TMS EEG power and TMS-evoked BOLD activity was computed per ROI for the three different ROI types. Subsequently, $p$-values were computed by contrasting the obtained rho values against null distributions of those values based on 20000 permutations of the TMS-evoked activity estimates using the Wilcoxon sign-rank test. This permutation procedure was also used in the second analysis, to compare the TMS-evoked hemodynamic activity between a "weak-power" and "strong-power" category (bin of 40 trials per category). Given the well-established negative correlations between spontaneous BOLD fluctuations and alpha power[13,14], the relation between alpha and BOLD responses was investigated with one-tailed permutation tests. All other statistical tests in this study were two-sided.

Results were considered significant at $p < 0.05$. Effect sizes ($d$; Cohen's D for dependent samples) indicate the standardized mean change as calculated by dividing the mean difference scores (between the evoked BOLD responses in the two categories) by the standard deviation of these difference scores. All statistical measures were computed using MATLAB (MathWorks, Natick, MA, USA). Note that analyses were performed at the single-subject level. Four participants each experienced between 120–150 TMS-events and 120–150 null-events (balanced per participant). In total, there were 525 TMS-events and 525 null-events (trials).

**Reporting summary**. Further information on research design is available in the Nature Research Reporting Summary linked to this article.

## Data availability
All source data underlying the figures (Figs. 1–4) are freely available via the open source repository Zenodo[74]. The complete datasets of the current study are available from the corresponding author on reasonable request. The data are stored in the data-repository of Maastricht University (Faculty of Psychology and Neuroscience).

## Code availability
Matlab code used to generate Figs. 3, 4 is available at open source repository Zenodo[74]. Spectral analyses of the EEG data were performed with the FieldTrip toolbox (source codes available via http://www.fieldtriptoolbox.org).

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

## Acknowledgements
The study received funding from the Netherlands Organization for Scientific Research (NWO: 453–15–008 AS, and VENI: 451–13–024 TdG) and the European Union's Seventh Framework Programme via the European Research Council (FP7/2007–2013/ERC AS) and the European FET Flagship project 'Human Brain Project' (FP7-ICT-2013-FET-F/604102 Grant Agreements, No. 7202070 (SGA1) and No. 785907 (SGA2) RG).

## Author contributions
Design of the experiment: J.C.P., J.R., A.T.S., R.G., T.S., T.d.G. Data collection: J.R., J.C.P., T.d.G., T.S., A.T.S. Providing material: R.G., A.T.S. Analysis and interpretation of the data: J.C.P., J.R., A.T.S., T.d.G. Drafting paper: J.C.P., J.R., T.d.G. Revising paper: J.C.P., J.R., A.T.S., R.G., T.S., T.d.G. Organization and supervision of the study: A.T.S.

## Competing interests
The authors declare no competing interests.
