## [Peer Review File · Communications Biology]

Reviewers' comments:

Reviewer #1 (Remarks to the Author):

Peters et al. present results from simultaneously combined EEG-TMS-fMRI measurements in $N = 4$ subjects, stimulating the right dorsal premotor cortex (dPMC) and evaluating the BOLD response to TMS in a set of motor related regions of interest as a function of pre-stimulus alpha power. They distinguish "high activators" ($N = 2$) and "low activators" ($N = 2$), with only the former showing proper motor system BOLD activation by a motor execution task (left finger movements) and TMS. They report lower TMS-evoked BOLD response in the motor system ROIs when TS was preceded by high alpha power in the "high activators".

The simultaneous combination of EEG, TMS, and fMRI is pioneering work, and I appreciate and encourage the efforts made by the authors. They have previously published a technical proof-of-feasibility paper (Peters et al., 2013) on this method. I do see the potential application and benefit from this extremely sophisticated and effortful technique, and I encourage the authors to further pursue this method development. However, the paper submitted here is not a technical paper (in fact it refers to their previous publication for most technical details), but is meant as an actual application to answer a specific neuroscientific question, namely the relevance of alpha oscillatory power for the excitability of a certain target region (here the dPMC) and the effective connectivity with other parts of the motor network. Unfortunately, the presented data do not extend beyond a mere proof-of-feasibility and are not suited to answer that question, given the substantial short-comings listed below.

1. The sample size of $N = 4$ (for the most crucial analyses even $N = 2$ "high activators") is not suited to answer human systems neuroscience questions regarding the role of alpha oscillations. Alpha oscillations and their link to cortical excitability show way too much variance both within and across subjects to derive any meaningful answer from such few subjects. Since two male and two female subjects are investigated it is even unclear whether the high/low activators are related to the sex of the participants.
2. "Low activators" were stimulated with lower TMS intensity, which is a clear confound with respect to the lower TMS-evoked activation.
3. The ROI approach (as opposed to whole brain analyses) of TMS-evoked BOLD responses and their EEG-dependent modulation buys sensitivity on the expense of specificity. That means, it is completely unclear whether TMS-evoked activations are limited to the motor system or whether they may show a completely unspecific activation pattern. Are other brain regions showing a completely different response or maybe just a very generic unspecific response to TMS?
4. Related to the point above, no sham stimulation has been performed to control for the massive auditory and somatosensory activation associated with TMS in an MR environment, and while the TMS-evoked patterns in bilateral M1 and SMA look plausible, other sensory activations (and their spread to the motor system) remain unclear.
5. The BOLD response related to the spontaneous fluctuation of alpha oscillations (which has been extensively studied with combined EEG-fMRI) has not been modelled in the fMRI analyses, and the BOLD signal changes owed to the low and high alpha power periods themselves are not taken into account when comparing the TMS-evoked BOLD responses during low and high alpha power periods.
6. Posterior (occipital and parietal) alpha oscillations are of particular strong SNR and amplitude and

can be measured in every single electrode on the head. The magnitude of posterior alpha power is much larger than that of the sensorimotor alpha (aka mu rhythm) and can easily mask it completely when not using local montages (Laplacian) or EEG source estimates. It can thus not be concluded at all, that the "local" EEG obtained from electrode FC4 and P2 reflects the local EEG signal. In fact, alpha is not particularly strong in the premotor cortex.

7. Given the investigation is focused on the motor system, the growing EEG-TMS literature on sensorimotor alpha should be considered.

8. Power of neighboring EEG bands (theta, gamma) has not been sufficiently investigated to justify statements like "[...] these additional analyses did reveal that the shunting of TMS-induced energy propagation by strong pre-TMS EEG power is limited to frequency bands below beta." as in line 234.

9. Phrasings like "[...] reflecting propagation of the inserted energy to structurally and functionally connected areas" (line 124) are very vague and I strongly suggest to use more neurophysiologically meaningful terms and interpretations.

Peters JC, Reithler J, Schuhmann T, de Graaf T, Uludag K, Goebel R, Sack AT (2013) On the feasibility of concurrent human TMS-EEG-fMRI measurements. *J Neurophysiol* 109:1214-1227.

Reviewer #2 (Remarks to the Author):

The mention that the investigators wished to avoid induction of peripheral motor activity is not explained. Stimulating at higher levels would seem to better parallel the active motor condition used to generate the stimulation targets and would be most relevant to the function of the motor system.

The choice of stimulation level only loosely based on motor threshold and adjusted for participant discomfort invalidates a categorization that the brain was effectively stimulated across participants. This confound carries over into the categorization of 'Low' vs. 'High Activators' that differed by stimulation intensity but also apparently by low activators having lower stimulation intensities relative to their actual motor thresholds due to scalp discomfort.

The EEG analyses seem to have been only conducted on the 'High Activators' which is a sample of only two participants. The TMS literature is consistent in showing high degrees of variability in TMS effects on brain and behavior that would suggest only two participants (or four) are not sufficient to characterize a TMS evoked response.

Minor points

The authors mention that the finger tapping and TMS evoked BOLD shape and time course were 'comparable.' It should be noted that the amplitudes are very different and that the finger tapping peak BOLD happens at the time of the lowest BOLD response in TMS evoked BOLD response.

Reviewer #3 (Remarks to the Author):

This is a methodological tour-de-force paper, integrating TMS (to stimulate premotor cortex), fMRI (to

track the propagation of TMS-induced activity across the large-scale brain network) and EEG (to study this signal propagation as a function of momentary oscillatory EEG state). The fMRI-results show that the spread of TMS-induced activity across the network depends on momentary alpha-amplitude (over premotor or parietal cortex) at TMS delivery, while beta-amplitude had no effect. This likely reflects trial-by-trial fluctuations in gating of signal from local to connected sites by modulations in oscillatory excitability states. A case is made for concurrent TMS-EEG-fMRI to open an exciting avenue for inferring the integrity of state-dependent network dynamics non-invasively, and the mechanisms of human network interactions, by combining the strength of the respective (dynamic and anatomical) brain imaging techniques.

This is a well-written paper. The methods are sound, the experiments performed to high standards (fMRI-localization, neuronavigation, avoiding peripheral motor confounds, controlled by EMG), the findings are new and the discussion interesting and well-taken (e.g. on a potential alpha/beta dissociation in the motor system). I have one main and a few minor points, which I hope can be addressed and will improve the paper.

Specific points

Main

1) The number of participants (n=4) is low. However, I don't see this as a problem here, given that the main emphasis is on the promise of the approach, and that the interpretation of alpha-activity as a modulator of gating is based on single-trial/-participant analysis (which substantiates the interest of using the approach to test network integrity in individuals/patients). However, where readers will take issue is with the presentation of data of only one participant (Figures 3 and 4 respectively). Given the low n, the results of each individual participants (at least high activators) should be shown in the main text. Plus: Is there any hint of differences between low and high alpha trials going in the right direction also in the low activators (participant 3 and 4), even if not significant, in which case it would be good to show these data in Figure form in the suppl Material.

Minor

2) The results reveal negative relationships between pre-TMS alpha-power and bold responses in selected fMRI ROIs (of the motor system). Would it be worthwhile to show that this is not observed for control-ROIs outside the motor system (e.g. over occipital cortex)?

3) Interpretation: The authors emphasize trial-by-trial variability in pre-TMS alpha activity (or the "meandering of sub-states of consciousness") to explain the results. However, there is another possible mechanism, which may explain the results. Alpha activity (over parietal cortex) shows slow drifts over an experiment, with these deterministic drifts (rather than stochastic variability) explaining the pre-stimulus/ response link (see Benwell et al., 2018, 2019). Could this alternative explanation of the results be considered, either in the discussion or by additional analysis? See Benwell et al. (2018, 2019) for approaches taking into considerations alpha increases with trial number.

Benwell CSY, London RE, Tagliabue CF, Veniero D, Gross J, Keitel C, Thut G. Frequency and power of human alpha oscillations drift systematically with time-on-task. *Neuroimage*. 2019 192:101-114.

Benwell CSY, Keitel C, Harvey M, Gross J, Thut G. Trial-by-trial co-variation of pre-stimulus EEG alpha power and visuospatial bias reflects a mixture of stochastic and deterministic effects. *Eur J Neurosci*. 2018 48(7):2566-2584.

4) Line 156; a bracket seems to be missing in front of (52%..

5) line 466: the average reference has been calculated based on a limited number of electrodes, which

is unusual. How have these electrodes been selected? Why is the selection of electrodes asymmetric (e.g. F3 not present) and why is FC4 (electrode of interest) not part of it while P2 (other electrode of interest) is? Please specify.

Response to reviewers

The changes in the manuscript and supplementary material that we made to improve our paper are clearly stated for each comment and referenced to by (p. xx).

Reviewers' comments:

Reviewer #1 (Remarks to the Author):

Peters et al. present results from simultaneously combined EEG-TMS-fMRI measurements in N = 4 subjects, stimulating the right dorsal premotor cortex (dPMC) and evaluating the BOLD response to TMS in a set of motor related regions of interest as a function of pre-stimulus alpha power. They distinguish 'high activators' (N = 2) and "low activators" (N = 2), with only the former showing proper motor system BOLD activation by a motor execution task (left finger movements) and TMS. They report lower TMS-evoked BOLD response in the motor system ROIs when TS was preceded by high alpha power in the "high activators".

The simultaneous combination of EEG, TMS, and fMRI is pioneering work, and I appreciate and encourage the efforts made by the authors. They have previously published a technical proof-of-feasibility paper (Peters et al., 2013) on this method. I do see the potential application and benefit from this extremely sophisticated and effortful technique, and I encourage the authors to further pursue this method development. However, the paper submitted here is not a technical paper (in fact it refers to their previous publication for most technical details), but is meant as an actual application to answer a specific neuroscientific question, namely the relevance of alpha oscillatory power for the excitability of a certain target region (here the dPMC) and the effective connectivity with other parts of the motor network. Unfortunately, the presented data do not extend beyond a mere proof-of-feasibility and are not suited to answer that question, given the substantial shortcomings listed below.

We want to thank the reviewer for acknowledging our efforts as pioneering work and for encouraging us to pursue this innovation of simultaneously combining EEG, TMS, and fMRI, which indeed is the result of many years of methodological development. We also greatly appreciate the reviewer's careful reading of our work. Below, we detail how we have followed many of the reviewer's excellent suggestions to substantially improve our manuscript. However, regarding this first (and related second) comment, we would like to stress that our main aim was to provide a first proof-of-principle demonstration of a meaningful application which critically depends on the presented trimodal set-up. By focusing on the link between alpha oscillatory power and motor network excitability as a showcase, we for the first time illustrate how concurrent EEG-TMS-fMRI enables the direct and noninvasive probing of oscillatory-brain-state dependent signal propagation within specific brain-wide functional networks. As such, the presented human subject data constitute an important exemplification of the trimodal approach's potential. Furthermore, we set out to assess whether signal gating, specifically modulating the impact of TMS throughout a wide network of brain regions, can be demonstrated on the single trial and single subject/patient level. It seems to us crucial that we managed to obtain these results in single subjects, for both the future research and the clinical applications we envision. This is exactly why we intentionally focused on single-subject and trial-by-trial analyses instead of group statistics, showcasing on single subject level how trial-by-trial pre-TMS EEG alpha and low-beta power fluctuations influenced TMS-induced signal propagation within the cortico-subcortical motor system. We consider this a strength and characteristic of our study. In this regard, we also respectfully refer to reviewer 3 who stated that the

sample size is not "a problem here, given that the main emphasis is on the promise of the approach, and that the interpretation of alpha-activity as a modulator of gating is based on single-trial/-participant analysis (which substantiates the interest of using the approach to test network integrity in individuals/patients)."

Compared to our previous technical feasibility study from 2013 to which the reviewer is also referring, we would like to point out the fundamental difference to our current manuscript. In our technical 2013 paper, we described how to establish a safe setup with adequate data quality, discussing results of a range of safety and MRI-sequence tests (including temperature measurements, stress safety tests, computations of magnetic field distortion, SNR-calculations, etc., primarily on a MR-phantom). Importantly, however, that previous study showed no proof of concept application of any kind, which is a very different issue. Hence, while our previous feasibility study showed that simultaneous EEG-fMRI-TMS can be safely implemented, the current study is the first proof-of-concept demonstration that it can yield meaningful results (unobtainable in any other way).

Kindly allow us to explain in just a little more detail why we are so excited about our findings. The current results demonstrate how concurrent TMS-EEG-fMRI uniquely allows to noninvasively chart the communication mechanisms within larger, dynamically interacting cortico-subcortical networks. We believe this to be an important result and message to various colleagues since the described approach provides a versatile framework for future studies on the propagation of TMS-induced activity with an unmatched high spatio-temporal resolution, allowing to test a wide range of related theories through flexible adaptation to the oscillatory band or TMS protocol of interest and its applicability to different network nodes with varying functional relevance. For example, TMS is increasingly used for therapeutic applications, e.g., for treating depression, or for supporting recovery processes and neurorehabilitation. It has long been known that while the direct stimulation effects of TMS are limited to superficial cortical brain sites, any cortical TMS stimulation always also will lead to indirect, remote effects in connected (sub)cortical structures. In fact, many behavioral effects in fundamental neuroscience research as well as therapeutic effects in the clinical applications of TMS depend on these indirect effects following the propagation of the cortical TMS signal within such cortico-subcortical networks. Moreover, the TMS impact is causally dependent on the state of neural excitability during application. Yet, until now it was impossible to control, modulate, or even properly assess the direction and strength of these remote, state-dependent network effects of TMS. Here, we demonstrate that simultaneous TMS-EEG-fMRI holds the potential to overcome these fundamental limitations, allowing us in future EEG-triggered TMS applications to apply TMS at predefined oscillatory brain states that either will or will not lead to specific remote brain activations by either facilitating or shunting signal propagation from the cortical TMS stimulation site. Relatedly, widespread research is underway that adapts rTMS protocols to align TMS pulses with real-time EEG power, or oscillatory phase, in different frequency bands, based on essentially untested assumptions that such oscillatory signatures directly impact the network connectivity and thereby efficacy of each TMS pulse. Our subject-level findings, presented here, provide important input for this rationale: TMS pulses throughout a network can indeed have larger effects depending on current oscillatory state. This demonstration alone should have great impact on the clinical direction the TMS field is investing in heavily. We look forward to using our TMS-fMRI-EEG approach to test for which (cognitive, emotional, etc) brain networks, and which oscillatory signatures in which frequency bands, this approach holds validity and promise. Our study opens the door to more efficient patient-customization of already globally applied rTMS treatments. But prior to any of that, it needed to be demonstrated that this approach can reveal meaningful results on the single-subject level in order to have future applications for patient-tailored TMS optimization.

Inspired by the reviewer's comment, we now substantially revised the manuscript to point out our main objectives much more clearly, highlight the focus on trial-by-trial and single-subject analyses, and in the discussion focus more on the potential of our findings, what they demonstrate, and their implications for neuroscience and personalized therapeutic neuromodulation. We hope that with these explanations and the shift in focus of the revised manuscript, the reviewer agrees with us (and reviewer 3) that the small sample size should not be reason for rejecting our work.

1. The sample size of $N = 4$ (for the most crucial analyses even $N = 2$ "high activators") is not suited to answer human systems neuroscience questions regarding the role of alpha oscillations. Alpha oscillations and their link to cortical excitability show way too much variance both within and across subjects to derive any meaningful answer from such few subjects. Since two male and two female subjects are investigated it is even unclear whether the high/low activators are related to the sex of the participants.

We fully agree with the reviewer about the need for larger samples when trying to answer new neuroscience questions at the population level, as well as the inter- and intra-individual variability in alpha oscillations. We therefore understand the reviewer's concern regarding the used sample size. However, we strongly believe that this concern does not apply to our current study design, and we kindly ask the reviewer to not reject our work based on the number of subjects measured. Please let us explain:

The question of a suitable, or statistically speaking, optimal sample size usually applies in studies aiming to provide group statistics and group inference testing. However, our main focus here was to first and foremost show that the employed trimodal combination of techniques can lead to meaningful and unique results (which has not been established previously), and second, that it can actually do so at the single-subject level, which ultimately is the most relevant perspective from a clinical point of view.

The reviewer rightfully points to the large variance in alpha oscillations observed in earlier studies. However, the large variance between trials and subjects is another reason that trial-by-trial and single-subject analyses are in fact stronger and more informative in our case. We show the single trial fluctuations to be systematically and significantly linked to specific network effects in a frequency specific manner. As such, the single-subject analyses presented here actually help to understand which factors contribute to the apparent variability in TMS effects encountered in the existing literature.

Finally, to comment on the reviewer's last point: sex was orthogonal to the high/low activators (one male, one female, for each). We now mention this in the revised manuscript and thank the reviewer for pointing out this oversight.

In sum, we honestly believe that our sample size was sufficient to address our research questions and provide the demonstrations we set out to achieve. Still, the reviewer might be left wondering why we did not simply add a few more subjects in light of his/her concern. So quite frankly: we are practically and technically not in the position to measure more subjects to increase the current sample. The used setup has undergone significant changes including a switch to a completely new MR scanner, a new flex MR coil, and a new MR compatible TMS system. Pooling data acquired with the current set up would be sub-optimal at best; rather we would need to fully start over. While we are investing to get this new set-up fully operational and are working on new pilots based on the proof-of-principle presented here, it would not be possible, or in our view meaningful, to increase the participant sample for the current manuscript, which as reviewer 3 also notes: already achieves our main goal of uniquely demonstrating "trial-by-trial fluctuations in gating of signal from local to connected sites by modulations in oscillatory excitability states". Many have assumed this mechanism to be at play. The current results show that our approach

can reveal it, for the first time. Showing these things again in more subjects would not seem to add so very much to this fundamental demonstration.

Ultimately, the most meaningful clinical applications of the current approach will also be based on single-subject measurements. As such, the fact that our results are observable in individual participants (and even that the approach might differentiate for which patients this does not work equally well) is, from a translational perspective, more relevant than a finding on group level. For instance, to optimize rTMS protocols to maximize plasticity modulation in individual patients for particular target brain networks, information and decisions will always have to be based on single-patient measurements.

2. “Low activators” were stimulated with lower TMS intensity, which is a clear confound with respect to the lower TMS-evoked activation.

Indeed, low activators were stimulated with lower TMS intensity, in terms of percentage of maximum stimulator output (MSO). In the revised manuscript we have acknowledged this point more explicitly and briefly discuss its implications, mirroring our reply below (p. 7).

Importantly, we do not consider this a problem for our main results/interpretations, for a number of reasons:

1. Our main results concern EEG state dependent modulation of fMRI responses in high activators, while keeping TMS intensity constant. A sufficiently high TMS-evoked BOLD response is required, to allow valid analyses of EEG state dependent modulations of such BOLD response. In low activators, TMS-evoked BOLD responses were too small (both in amplitude as well as the spatial extent of the response), to allow a valid analysis of its EEG-driven modulation.
2. We do not interpret differences between ‘low activator brains’ and ‘high activator brains’. As such, there is nothing to confound, since we did not intend to draw any conclusions on the strength of BOLD responses in general. We only identified that half of our participants had very low BOLD responses in the motor system to begin with, leaving the other half for meaningful further analysis. This is a methodological consideration, not an interpretative issue. We certainly did not intend to conclude that EEG state dependence is a mechanism which only applies to ‘high activator brains’, for example, and have checked that the revised manuscript does not imply such conclusions.
3. ‘Low activators’ had a proclivity for low fMRI activity of the motor system as observed in the localizer fMRI experiment (i.e., completely independent of any TMS application): Several ROIs could not be defined in low activators, due to a lack of significant activation. Moreover, the subset of ROIs that was identified in low activators consistently encompassed substantially fewer voxels compared to the same regions in high activators. As such, low activators were not characterized necessarily by low TMS-evoked BOLD activations, but low BOLD activations in the motor system generally.
4. This does not exclude the possibility that, as the reviewer points out, stimulating at lower TMS intensity could have contributed to the small TMS-evoked BOLD responses in low activators. But this is also not necessarily the case. We determined TMS intensity for each participant based on individual motor threshold (as determined using single TMS pulses over M1), rather than maximum stimulator output, in line with several other dorsal premotor TMS studies (e.g., Lee et al., 2006). While TMS intensity in terms of MSO was lower for low activator participants, this did not translate directly to TMS intensity in terms of percentage MT, which is assumed to capture the functional impact of TMS at the level of the cortex: For one low activator, the MT-relative intensity was indeed lower to accommodate discomfort, but for the other, it was a little higher than in the high activators to accommodate our lower limit on MSO. These specifics have been further clarified in the revised method section (p. 16).

In the revised manuscript, we discuss on pages 7 and 8 that both an idiosyncratic low hemodynamic activation of the motor system, as well as the lower TMS intensity, might have contributed to the lower TMS-evoked activity in low activators, and that the lower TMS intensity could have obscured any EEG-driven effects in the low activators. Moreover, the manuscript elaborates on the lack of effects in a subset of our subject sample, which actually echoes several reports in the TMS literature (e.g., Diekhoff-Krebs, et al., 2017), and discusses how future work involving the trimodal set-up exemplified in the current study might help to elucidate such discrepancies further (on p. 13 of the revised manuscript).

3. The ROI approach (as opposed to whole brain analyses) of TMS-evoked BOLD responses and their EEG-dependent modulation buys sensitivity on the expense of specificity. That means, it is completely unclear whether TMS-evoked activations are limited to the motor system or whether they may show a completely unspecific activation pattern. Are other brain regions showing a completely different response or maybe just a very generic unspecific response to TMS?

This is indeed important to determine. Inspired by this comment, we decided to pursue this concrete suggestion and conducted additional whole-brain analyses and new (non-motor) ROI analyses:

1. *New whole-brain analyses: For each participant, a whole-brain GLM map was created based on a design matrix comprising 3 box-car predictors (strong, medium, weak parietal alpha power trials) convolved with a two-gamma hemodynamic response function (HRF) and head motion (3) and rotation (3) confound predictors to correct for potential influences of head motion. These maps did not show any meaningful activation patterns in extra-motor areas (with p-values adjusted for multiple comparisons with a False Discovery Rate correction, using a proportion of false discoveries of $q = 0.05$; Benjamini and Hochberg, 1995).*
2. *Additional non-motor ROI analyses: Furthermore, to increase specificity, we defined additional Non-Motor ROIs (e.g., primary auditory cortex). These new results (reported in the revised manuscript, p. 10), show that our findings were indeed non-generic and largely specific to the motor system network. Of course, our research question specifically concerned the relationship of pre-TMS EEG and post-TMS BOLD responses within the motor system, following related work on the motor system of Sauseng et al. (2009) and Romei et al. (2016). The absence of significant effects in some Motor-Execution VOIs (e.g., left Caudate Nucleus and left PMd) appeared to suggest that our main observations were not a completely unspecific pattern already throughout the motor system. However, the new analyses prompted by the reviewer nicely complete the overview of EEG-state specific TMS effects.*

4. Related to the point above, no sham stimulation has been performed to control for the massive auditory and somatosensory activation associated with TMS in an MR environment, and while the TMS-evoked patterns in bilateral M1 and SMA look plausible, other sensory activations (and their spread to the motor system) remain unclear.

Indeed, this comment reflects a perennial problem in simultaneous TMS-fMRI, lacking a perfect solution to date. That said, we have determined (see previous comment/response and the newly added analyses) that ROIs outside the targeted motor network either do not display the same pattern of responses or do so in a much less pronounced manner. Furthermore, auditory stimulation by itself might be hardly effective in driving our main areas of interest, as suggested by e.g. the flatline response in PMd to the 'audio cueing' control condition of the Motor-Execution fMRI localizer session (visible in figure 1A).

Still, a perfect sham control seems not to exist for TMS-fMRI, and previous TMS-fMRI studies therefore simply decreased TMS intensity as their main control condition. Although we cannot fully exclude all alternative explanations, we would suggest that both the intrinsic nature of the design, and the pattern of results, suggest that we have actually achieved a new level of sophistication and unambiguity of findings, relative to previous TMS-fMRI studies, because: 1) The core analysis focuses on the different magnitude of motor network BOLD responses depending on EEG state while the TMS intensity was kept constant – therefore, the associated feedforward auditory/somatosensory responses should be similar between the core experimental conditions (i.e., low alpha vs. high alpha power states), 2) though post-hoc, we had two participants with the hypothesized pattern of results on trial-by-trial/subject-level, and two participants with less strong results, even though the auditory clicks produced by the TMS coil were clearly audible for all subjects. And although this did consequently lead to BOLD activations in auditory cortex, none of the low activators actually showed a significant modulation based on alpha power, suggesting the auditory stimulation by itself is not likely to drive the observed effects. These arguments are not conclusive, but it is uncertain whether the addition of TMS to a different target site (with inevitably different somatosensory and auditory effects) would offer better control than the intrinsic control of demonstrating state-dependence of BOLD responses specifically in a priori regions-of-interest. Most importantly therefore: 3) especially now that this is bolstered by the reviewer-suggested additional analyses (see previous points), the regions in which we observe the hypothesized pattern of results are exactly those of interest (i.e. motor network), where we would not expect such modulations of BOLD responses if caused by nonspecific TMS confounders (see also next point).

5. The BOLD response related to the spontaneous fluctuation of alpha oscillations (which has been extensively studied with combined EEG-fMRI) has not been modelled in the fMRI analyses, and the BOLD signal changes owed to the low and high alpha power periods themselves are not taken into account when comparing the TMS-evoked BOLD responses during low and high alpha power periods.

Our research objective concerned TMS-evoked BOLD responses rather than spontaneous fMRI fluctuations, which indeed have been a main interest in several EEG-fMRI studies. Our analysis methods are therefore mainly inspired by TMS-fMRI and TMS-EEG studies, which investigate the propagation of activity induced by TMS stimulation. That said, we appreciate the interesting suggestion of the reviewer and performed additional analyses to model the relation between EEG power and spontaneous BOLD activity. Specifically, we analysed so-called 'Null-events', which were presented in interleaved fashion with TMS-events but did not include any actual TMS pulse delivery. Null-events were matched to TMS-events in terms of their number of occurrences and followed a similar temporal distribution. In this way, we could establish that relatively few of our main ROIs showed an influence of ongoing alpha power on spontaneous hemodynamic activity (7/35 ROIs = 20% for the correlation analyses; 13/35 ROIs = 37.1% for the category analyses). So even though some spontaneous coupling was observed (in line with the mentioned EEG-fMRI studies), the reported relationship between ongoing alpha-power and the ensuing BOLD amplitude following actual TMS events was found to be much more pronounced, affecting 2 to 3 times more ROIs (77.14% and 88.6% for the correlation and category analyses, respectively). In addition, a novel set of analyses on non-Motor control ROIs corroborated these results, indicating that areas not directly connected to the TMS target site do not mimic the alpha power-driven modulations of TMS-evoked responses in motor areas. Together, these new analyses show that the observed alpha modulation of TMS-evoked propagation patterns is not simply a reflection of ongoing covariations between alpha power and spontaneous BOLD fluctuations. These new results have now been included in the revised manuscript (p. 10).

6. Posterior (occipital and parietal) alpha oscillations are of particular strong SNR and amplitude and can be measured in every single electrode on the head. The magnitude of posterior alpha power is much larger than that of the sensorimotor alpha (aka mu rhythm) and can easily mask it completely when not using local montages (Laplacian) or EEG source estimates. It can thus not be concluded at all, that the “local” EEG obtained from electrode FC4 and P2 reflects the local EEG signal. In fact, alpha is not particularly strong in the premotor cortex.

We appreciate this comment. The relation between FC4 alpha and beta power and BOLD activity was merely investigated to explore whether EEG power at the TMS target site influenced TMS effects (and was therefore implemented as ancillary result in the Supplementary Material instead of the main manuscript). We agree with the reviewer and realize, in hindsight, that given the strong dominance of parietal alpha (and its relatively close proximity to the FC4 electrode), truly local alpha activity cannot be unambiguously distinguished from parietal alpha influences. To accommodate the reviewer, and avoid any confusions for the reader, we have removed the FC4 alpha analyses from the revised manuscript.

7. Given the investigation is focused on the motor system, the growing EEG-TMS literature on sensorimotor alpha should be considered.

We thank the reviewer for this suggestion. We have now added a new paragraph in the revised discussion, which reviews EEG-TMS studies on sensorimotor alpha (p. 11).

8. Power of neighboring EEG bands (theta, gamma) has not been sufficiently investigated to justify statements like “[...] these additional analyses did reveal that the shunting of TMS-induced energy propagation by strong pre-TMS EEG power is limited to frequency bands below beta.” as in line 234.

We have removed this sentence in the revised manuscript (p. 10).

9. Phrasings like “[...] reflecting propagation of the inserted energy to structurally and functionally connected areas” (line 124) are very vague and I strongly suggest to use more neurophysiologically meaningful terms and interpretations.

We appreciate the reviewer pointed us to these unclear phrasings in our manuscript. Line 124 (now line 154-155) and various other places throughout the revised manuscript have been rephrased accordingly.

References

- Benjamini, Y., Hochberg, Y., 1995. Controlling the False Discovery Rate - a Practical and Powerful Approach to Multiple Testing. *Journal of the Royal Statistical Society Series B-Statistical Methodology* 57, 289-300.
- Diekhoff-Krebs, S. et al. Interindividual differences in motor network connectivity and behavioral response to iTBS in stroke patients. *NeuroImage Clin.* 15, 559–571 (2017).
- Romei, V. et al. Causal evidence that intrinsic beta-frequency is relevant for enhanced signal propagation in the motor system as shown through rhythmic TMS. *Neuroimage* 126, 120–130 (2016).
- Sauseng, P., Klimesch, W., Gerloff, C. & Hummel, F. C. Spontaneous locally restricted EEG alpha activity determines cortical excitability in the motor cortex. *Neuropsychologia* 47, 284–288 (2009).

#####

Reviewer #2 (Remarks to the Author):

The mention that the investigators wished to avoid induction of peripheral motor activity is not explained. Stimulating at higher levels would seem to better parallel the active motor condition used to generate the stimulation targets and would be most relevant to the function of the motor system.

We thank the reviewer for this perceptive remark. Indeed, applying TMS at such high intensity that muscle twitches would be induced, would more explicitly mimic the active motor condition. For many research questions this would be the preferred approach. However, we explicitly wished to avoid this, due to the aims of the current study. We agree this was not well explained in our original manuscript, so on pages 4 and 6 we clarified our rationale: The current research intended to demonstrate that, on the trial-by-trial and single-subject level, signal propagation is EEG state dependent. TMS drives activity in one motor system node, and depending on EEG state, the BOLD response throughout the motor network is modulated. The active motor condition was included primarily to define the precise location of key areas within the motor system of each participant individually, to serve as ROIs for the subsequent TMS-based analyses. As such, we did not actually want to mimic this active motor condition.

In fact, early TMS-fMRI studies (Bestmann et al., 2003, 2004) found that supra-motorthreshold TMS induces less interpretable BOLD responses in the motor system, exactly because of the induced muscle twitch. We were interested in mapping TMS-evoked efferent responses throughout the motor system. A TMS-evoked peripheral motor response would have also caused reafferent processing in the motor system, thereby actually creating a sensorimotor confound (Bestmann et al. 2003, 2004). Stimulating at non-primary motor regions (Bestmann et al., 2005) and at a sufficiently low intensity to avoid peripheral motor activity, ensures that any observed TMS-induced BOLD responses throughout the motor network can only be ascribed to forward signal propagation from the TMS site to the rest of the network, not to reafferent signals caused by an induced muscular response.

We wanted to assess specifically the gating of forward signal propagation, without such confounding sensorimotor activations. This research goal trumped matching as closely as possible the active motor condition. We have clarified both these goals (p 4) and made explicit this rationale for avoiding induction of peripheral motor activity (p 6).

The choice of stimulation level only loosely based on motor threshold and adjusted for participant discomfort invalidates a categorization that the brain was effectively stimulated across participants. This confound carries over into the categorization of 'Low' vs. 'High Activators' that differed by stimulation intensity but also apparently by low activators having lower stimulation intensities relative to their actual motor thresholds due to scalp discomfort.

In hindsight, we realize our concise description on the selected stimulation intensity might have led to this impression. Based on the reviewer's comment, we have further explained our rationale and discussed implications regarding stimulation differences as outlined below, in the revised manuscript (p. 7).

Please allow us to clarify the rationale behind the choice of stimulation intensity and its implications:

*In line with several other dorsal premotor TMS studies (e.g., Bestmann et al., 2010; Lee et al., 2006), we determined TMS intensity for each participant based on individual motor threshold (as determined using single TMS pulses over M1), which captures the functional impact of TMS at the level of the cortex. Note that unlike other single-pulse dorsal premotor TMS studies, we use a potent **triple-pulse***

TMS protocol (i.e., 3 pulses at 15 Hz), so we had to ensure stimulation was not too strong (i.e., inducing peripheral motor activity) rather than too weak.

For (only) one participant (a low activator) we had to slightly reduce TMS intensity due to subject discomfort. However, this does not translate directly to having an ineffective stimulation: For the other low activator, the intensity was a little higher than in the high activators (in terms of percentage of rMT) to accommodate our lower limit on maximum stimulator output (45%). The latter indicates that despite a relatively high TMS stimulation (relative to MT and thus stimulation in terms of functional effects), TMS-evoked BOLD responses can be relatively low. The low TMS-evoked fMRI activity in low activators might be related to a general difference in BOLD activity in high and low activators rather than TMS intensity: 'Low activators' had a proclivity for low fMRI activity of the motor system as observed in the localizer fMRI experiment (i.e., completely independent of any TMS application): Several ROIs could not be defined in low activators, due to a lack of significant activation. Moreover, the subset of ROIs that was identified in low activators consistently encompassed substantially fewer voxels compared to the same regions in high activators. As such, low activators were not characterized necessarily by low TMS-evoked BOLD activations, but by their low BOLD activations in the motor system in general.

On pages 7 and 8 of the revised manuscript, we discuss that both an idiosyncratic low hemodynamic activation of the motor system, as well as the lower TMS intensity, might have contributed to the lower TMS-evoked activity in low activators, and acknowledge that the lower TMS intensity could have obscured any EEG-driven effects in the low activators.

Most importantly, differences in stimulation intensity per se do not affect the interpretation of our findings: Our main results concern EEG state dependent modulation of fMRI responses in high activators, while keeping TMS intensity constant. We do neither interpret differences between 'low activator brains' and 'high activator brains', nor the strength of BOLD responses in general. We only identified that half of our participants had very low BOLD responses in the motor system to begin with, leaving the other half for meaningful further analysis. Thus, this is a methodological consideration (subset selection), not an interpretative issue (related to potential confounds). We certainly did not intend to conclude that EEG state dependence is a mechanism which only applies to 'high activator brains', for example, and have checked that the revised manuscript does not imply such conclusions. Moreover, the revised manuscript elaborates on the lack of effects in a subset of our subject sample, which actually echoes several reports in the TMS literature (e.g., Diekhoff-Krebs, et al., 2017) and how future work involving the trimodal set-up exemplified in the current study might help to elucidate such discrepancies further (p. 13).

The EEG analyses seem to have been only conducted on the 'High Activators' which is a sample of only two participants. The TMS literature is consistent in showing high degrees of variability in TMS effects on brain and behavior that would suggest only two participants (or four) are not sufficient to characterize a TMS evoked response.

Our study is meant as a first demonstration of how concurrent EEG-TMS-fMRI enables the direct and noninvasive probing of brain-state dependent signal propagation within specific brain-wide functional networks, in this case the cortico-subcortical motor network. Our goal was thus not primarily to establish alpha power gating of signal propagation as a new neuroscientific finding on the group level, generalizable to the population. This would indeed require a larger sample size and different (group) statistics. Rather, we assessed whether signal gating, specifically modulating the efficacy of TMS, can be demonstrated on the single trial and single subject/patient level. It seems to us crucial that we managed to obtain these results in single subjects, for both the future research and the clinical applications we envision. This is exactly why we intentionally focused on single-subject and trial-by-trial analyses instead of group

statistics, showcasing on single subject level how trial-by-trial pre-TMS EEG alpha and low-beta power fluctuations influenced TMS-induced signal propagation within the cortico-subcortical motor system. We consider this a strength and characteristic of our study. With this regard, we also respectfully refer to reviewer 3 who also stated that the sample size is not "a problem here, given that the main emphasis is on the promise of the approach, and that the interpretation of alpha-activity as a modulator of gating is based on single-trial/-participant analysis (which substantiates the interest of using the approach to test network integrity in individuals/patients)."

The reviewer rightfully points to the large variance in TMS effects. However, the large variance between trials and subjects is another reason that trial-by-trial and single-subject analyses are in fact stronger and informative in our case. We show the single trial fluctuations to be systematically and significantly linked to specific network effects in a frequency-dependent manner. As such, the single-subject analyses presented here actually help to understand which factors contribute to the apparent variability in TMS effects encountered in the existing literature.

In sum, we honestly believe that our sample size was sufficient to address our research questions and provide the demonstrations we set out to achieve. Ultimately, the most meaningful clinical applications of the current approach will also be based on single-subject measurements. As such, the fact that our results are observable in individual participants (and even that the approach might differentiate for which patients this does not work equally well) is, from a translational perspective, much more relevant than a finding on group level. For instance, to optimize rTMS protocols to maximize plasticity modulation in individual patients for particular target brain networks, information and decisions will always have to be based on single-patient measurements.

Minor points

The authors mention that the finger tapping and TMS evoked BOLD shape and time course were 'comparable.' It should be noted that the amplitudes are very different and that the finger tapping peak BOLD happens at the time of the lowest BOLD response in TMS evoked BOLD response.

We thank the reviewer for pointing this out. Indeed, 'comparable' was too crudely phrased. We rather meant to say that the TMS-induced BOLD response has the hypothesized shape, timing, and amplitude, when taking into account the finger tapping response and correcting the timing (should be earlier) and amplitude (should be smaller) of the BOLD peaks. These differences are according to hypothesis since TMS at time 0 of the evoked response function directly induces cortical action potentials and the TMS event is of shorter duration (i.e., 133 ms) than the motor execution block (i.e., repeated movements throughout a 4 second period). This may moreover relate to the lower amplitude itself, which is expected since an actual motor action (finger tapping) both involves more extensive cortical activation than that induced by a TMS pulse, and includes the sensorimotor feedback component discussed above to an earlier concern raised by this reviewer.

We have rephrased this remark in the revised manuscript, removing the confusing mention that the BOLD responses are comparable and rather indicating that the TMS induced BOLD response is as expected given the finger tapping BOLD response but taking into account the difference between a voluntary motor action and a subthreshold TMS pulse (text p. 6 and legend Figure 1).

References

- Bestmann, S., Baudewig, J., Siebner, H. R., Rothwell, J. C., & Frahm, J. (2003). Subthreshold high-frequency TMS of human primary motor cortex modulates interconnected frontal motor areas as detected by interleaved fMRI-TMS. *NeuroImage*, 20(3), 1685–1696.
- Bestmann, S., Baudewig, J., Siebner, H. R., Rothwell, J. C., & Frahm, J. (2004). Functional MRI of the immediate impact of transcranial magnetic stimulation on cortical and subcortical motor circuits. *European Journal of Neuroscience*, 19(7), 1950–1962.
- Bestmann, S., Baudewig, J., Siebner, H. R., Rothwell, J. C., & Frahm, J. (2005). BOLD MRI responses to repetitive TMS over human dorsal premotor cortex. *NeuroImage*, 28(1), 22–29.
- Bestmann, S. et al. (2010). The Role of Contralesional Dorsal Premotor Cortex after Stroke as Studied with Concurrent TMS-fMRI. *J. Neurosci.* 30, 11926–11937
- Diekhoff-Krebs, S. et al. Interindividual differences in motor network connectivity and behavioral response to iTBS in stroke patients. *NeuroImage Clin.* 15, 559–571 (2017).
- Lee, J.H, van Donkelaar, P. (2006). The human dorsal premotor cortex generates on-line error corrections during sensorimotor adaptation. *J. Neurosci.* 26, 3330 –3334

Reviewer #3 (Remarks to the Author):

This is a methodological tour-de-force paper, integrating TMS (to stimulate premotor cortex), fMRI (to track the propagation of TMS-induced activity across the large-scale brain network) and EEG (to study this signal propagation as a function of momentary oscillatory EEG state). The fMRI-results show that the spread of TMS-induced activity across the network depends on momentary alpha-amplitude (over premotor or parietal cortex) at TMS delivery, while beta-amplitude had no effect. This likely reflects trial-by-trial fluctuations in gating of signal from local to connected sites by modulations in oscillatory excitability states. A case is made for concurrent TMS-EEG-fMRI to open an exciting avenue for inferring the integrity of state-dependent network dynamics non-invasively, and the mechanisms of human network interactions, by combining the strength of the respective (dynamic and anatomical) brain imaging techniques.

This is a well-written paper. The methods are sound, the experiments performed to high standards (fMRI-localization, neuronavigation, avoiding peripheral motor confounds, controlled by EMG), the findings are new and the discussion interesting and well-taken (e.g on a potential alpha/beta dissociation in the motor system). I have one main and a few minor points, which I hope can be addressed and will improve the paper.

We greatly appreciate that the reviewer regards our work novel, scientifically sound, and following high standards. Moreover, we thank the reviewer for considering our paper well-written, a methodological tour-de-force, with an interesting and well-taken discussion. We have implemented all the excellent suggestions by the reviewer, as detailed below point-by-point, which has clearly further improved the manuscript.

Specific points

Main

1) The number of participants (n=4) is low. However, I don't see this as a problem here, given that the main emphasis is on the promise of the approach, and that the interpretation of alpha-activity as a modulator of gating is based on single-trial/-participant analysis (which substantiates the interest of using

the approach to test network integrity in individuals/patients). However, where readers will take issue is with the presentation of data of only one participant (Figures 3 and 4 respectively). Given the low n, the results of each individual participants (at least high activators) should be shown in the main text. Plus: Is there any hint of differences between low and high alpha trials going in the right direction also in the low activators (participant 3 and 4), even if not significant, in which case it would be good to show these data in Figure form in the suppl Material.

Indeed, the main focus of this paper is on the promise of this novel approach, which is particularly suited to test network integrity in individuals and particularly in patients, following current developments in the NIBS field. Therefore, we have further clarified in the revised manuscript (page 7) that the high activators were a sub-sample selected for further analyses, given our goal to present single-subject results as a first proof-of-principle demonstration of this new research approach. In order to clearly point out this rationale and maintain consistency across all reported main analyses (which solely focus on high activators), we prefer not to include figures reporting fMRI differences between low and high alpha trials for low activators, as we believe this would reduce clarity. However, if the reviewer prefers otherwise, we can include these figures in a further revision.

Following the reviewer's helpful suggestion, we have now included the results from all high activators in Figure 3 and 4. We have also reformatted both figures and Table 1 to consistently show the data from the same participant first (hence the updated mesh reconstructions in the top panel of Figure 3).

Minor

2) The results reveal negative relationships between pre-TMS alpha-power and bold responses in selected fMRI ROIs (of the motor system). Would it be worthwhile to show that this is not observed for control-ROIs outside the motor system (e.g. over occipital cortex)?

We thank the reviewer for this excellent suggestion. New analyses have been performed on control ROIs outside the motor system (see reviewer 1 points 3 and 4 for details). These complementary results confirm that the TMS effects in the motor system are non-generic as the number of ROIs which show the described alpha modulation is consistently higher within as compared to outside the targeted motor network. We agree this was important to assess and the revised manuscript discusses the corresponding analyses (p. 10).

3) Interpretation: The authors emphasize trial-by-trial variability in pre-TMS alpha activity (or the “meandering of sub-states of consciousness”) to explain the results. However, there is another possible mechanism, which may explain the results. Alpha activity (over parietal cortex) shows slow drifts over an experiment, with these deterministic drifts (rather than stochastic variability) explaining the pre-stimulus/response link (see Benwell et al., 2018, 2019). Could this alternative explanation of the results be considered, either in the discussion or by additional analysis? See Benwell et al. (2018, 2019) for approaches taking into considerations alpha increases with trial number.

We thank the reviewer for pointing out this possibility. Adopting the approach outlined in the paper by Benwell et al. (2019), we performed additional analyses to estimate a drift in alpha power across trials. We did not observe such a drift when assessing the correlation between alpha power and trial number (all p 's > 0.05). One reason for this discrepancy might be that in the studies referred to by the reviewer, slow drifts in alpha power were investigated within the context of a behavioral task involving many trial repetitions (even though the analyses were largely limited to the pre-stimulus time window). This could

have led to increases in fatigue or a loss of focus towards the end of the measurement session. In contrast, in our experiment participants were not engaged in any demanding behavioral task (and runs had a relatively short duration), possibly leading to a more homogeneous state in terms of mental effort.

That said, the issue of potential drifts is nevertheless important to address (also when considering optimizing the efficacy of clinical applications of TMS). Therefore, we have incorporated these additional analyses and results in the revised manuscript, and also explained their relevance in the light of this alternative mechanism involving deterministic drifts (p. 19).

4) Line 156; a bracket seems to be missing in front of (52%..

We appreciate the reviewer's careful reading and have corrected this omission in the revised manuscript (p. 7).

5) line 466: the average reference has been calculated based on a limited number of electrodes, which is unusual. How have these electrodes been selected? Why is the selection of electrodes asymmetric (e.g. F3 not present) and why is FC4 (electrode of interest) not part of it while P2 (other electrode of interest) is? Please specify.

In the used average reference, we did not include temporal, occipital and the most inferior and most frontal electrodes. This was due to the specifics of our setup: the EEG signal in some of these electrodes was relatively noisy due to varying sources (e.g., poor cap fit, subject movement, muscle artifacts).

Regarding the second question: We are grateful to the reviewer for pointing us to this important typo in the manuscript: we meant FC4 instead of F4. That is, FC4 was part of the average reference and F3 and its symmetric counterpart F4 were both not included. We apologize for any confusion and have corrected this typo in the revision (p. 18). The remark of the reviewer inspired us to reconsider our selection beyond ensuring a left-right symmetry. In order to increase anterior-posterior symmetry of the average reference (i.e., the number of frontal versus central and parietal electrodes), we have added a few frontal electrodes (e.g., F3 and F4) to the average reference and reran the analyses. This change in average reference did not qualitatively affect our results. The new results are implemented in the revised manuscript (results section pp. 7-10 and pp. 18-19).

REVIEWERS' COMMENTS:

Reviewer #1 (Remarks to the Author):

I thank the authors for their extensive response to my concerns. I understand that the paper was more intended as a proof-of-principle for the TMS-EEG-fMRI application, and I acknowledge that this was now made more transparent in the revised version of the manuscript. I can follow the majority of arguments made by the authors to defend their approach and the validity of the results even for a sample size of $N = 4$. I am sufficiently satisfied by the revision and will not further insist on any of my previously raised concerns. I wish to state once more that I very much appreciate and encourage the authors endeavor to simultaneously combine TMS, EEG, and fMRI, which sets the stage for many interesting research applications.

Reviewer #2 (Remarks to the Author):

The authors tempered a number of problematic statements, listed additional caveats to their results, and generally were responsive to my comments. One of my comments, however, was only partially addressed:

"The EEG analyses seem to have been only conducted on the 'High Activators' which is a sample of only two participants. The TMS literature is consistent in showing high degrees of variability in TMS effects on brain and behavior that would suggest only two participants (or four) are not sufficient to characterize a TMS evoked response."

The authors explain that a focus on only two participants is reasonable because they want to be able to describe single subject/patient effects and they go on to mention "clinical applications we envision." However, they do not address the focus only on High Activators mentioned in my comment. I believe this to be an important major caveat to their findings especially in terms of clinical relevance to individual patients. I believe that the authors need to qualify their findings "for a very specific cohort (high activators), there MAY be some ultimate clinical use" or similar to clearly indicate that the findings should not be expected generalize to a typical patient in a clinical setting.

Reviewer #3 (Remarks to the Author):

The authors have addressed all my comments, but with the revisions a few additional (minor to moderate) points have surfaced.

Specific points

1) Presenting single-participant data:

Given the sample size of this study, the authors now show all single participant data in Figures 3 and 4 (which is good) but I think this should also be applied to Figure 2 on ROIs, unless these have been identified on the group level, not the individual level. If the latter is the case, this should be made clearer in the main text. Also, on line 166, the sentence "the same subset of motor regions (when present) was defined by .." is confusing, as the two ways of defining ROIs do not lead to exactly the same outcome. I would reformulate to "a further subset of motor regions (with partial overlap with the first set, see below) was defined by .."

2) Figure 4: Why are data (boxes) missing in a subset of the panels/ conditions? Could this be clarified in the legend and/or main text?

3) Additional analyses to probe for specificity of the frontal alpha dependent BOLD responses in the motor system.

3.1.) The authors have tested a further set of ROIs to show that frontal alpha does predict to a lesser extent the BOLD responses in areas outside the motor system, which is important (I count 3 control areas x 2 sides = 6?). However, I would recommend to reconsider how this data is presented (a bit messy as is, lines 252 - 259). Could the results be summarized in one sentence simply stating the percentage of significant correlations/ comparisons (as done in the sentence above on null-events)?

3.2.) It would seem important as well to test the specificity to frontal alpha by examining how a non-frontal EEG alpha signal would predict BOLD responses in motor areas. The parietal alpha signal (that has already been extracted, lines 546-547) could serve for this purpose.

4) Does Suppl Fig 4 reflect activity over FC4?

5) There are a few issues with sentence constructs/ language, e.g. line 156. Please check.

Response to reviewers

The changes in the manuscript and supplementary material that we made to improve our paper are clearly stated for each comment and referenced to by (p. xx).

Reviewer #1 (Remarks to the Author):

I thank the authors for their extensive response to my concerns. I understand that the paper was more intended as a proof-of-principle for the TMS-EEG-fMRI application, and I acknowledge that this was now made more transparent in the revised version of the manuscript. I can follow the majority of arguments made by the authors to defend their approach and the validity of the results even for a sample size of $N = 4$. I am sufficiently satisfied by the revision and will not further insist on any of my previously raised concerns. I wish to state once more that I very much appreciate and encourage the authors endeavor to simultaneously combine TMS, EEG, and fMRI, which sets the stage for many interesting research applications.

We thank the reviewer for his/her benevolent response. We greatly appreciate the acknowledgment and encouragement regarding our efforts and want to thank him/her again for the careful consideration of our manuscript.

Reviewer #2 (Remarks to the Author):

The authors tempered a number of problematic statements, listed additional caveats to their results, and generally were responsive to my comments. One of my comments, however, was only partially addressed:

"The EEG analyses seem to have been only conducted on the 'High Activators' which is a sample of only two participants. The TMS literature is consistent in showing high degrees of variability in TMS effects on brain and behavior that would suggest only two participants (or four) are not sufficient to characterize a TMS evoked response."

The authors explain that a focus on only two participants is reasonable because they want to be able to describe single subject/patient effects and they go on to mention "clinical applications we envision." However, they do not address the focus only on High Activators mentioned in my comment. I believe this to be an important major caveat to their findings especially in terms of clinical relevance to individual patients. I believe that the authors need to qualify their findings "for a very specific cohort (high activators), there MAY be some ultimate clinical use" or similar to clearly indicate that the findings should not be expected generalize to a typical patient in a clinical setting.

We agree with the reviewer that it is important to realize that targeted TMS effects often only occur in part of the sample (in our study, but also in many other studies of which some are discussed in the manuscript). In our case, we could identify these participants ("Low Activators") by an idiosyncratic low hemodynamic activation of the motor system, as observed in the localizer fMRI experiment (i.e., completely independent of any TMS application). Future work could explore whether combined TMS-fMRI(-EEG) might help to differentiate for which patients therapeutic effects might be more difficult to obtain (e.g., those with a proclivity for low activation which appears to reduce remote TMS effects). The reviewer is thus completely right that our results, like all other TMS findings, should not be generalized to

be of clinical relevance for every single patient. Instead, the here presented methodology could be used for stratification, in helping to identify which specific subgroup of patients may benefit clinically.

Following the reviewer's suggestion, we now mention that 1) our findings cannot be generalized to all patients and that 2) in the clinical setting it is important to carefully select suitable candidates for therapeutic TMS interventions and the proposed approach allows to further investigate which patients might benefit most (p. 11).

Reviewer #3 (Remarks to the Author):

The authors have addressed all my comments, but with the revisions a few additional (minor to moderate) points have surfaced.

Specific points

1) Presenting single-participant data:

Given the sample size of this study, the authors now show all single participant data in Figures 3 and 4 (which is good) but I think this should also be applied to Figure 2 on ROIs, unless these have been identified on the group level, not the individual level. If the latter is the case, this should be made clearer in the main text. Also, on line 166, the sentence "the same subset of motor regions (when present) was defined by .." is confusing, as the two ways of defining ROIs do not lead to exactly the same outcome. I would reformulate to "a further subset of motor regions (with partial overlap with the first set, see below) was defined by .."

We appreciate the reviewer's comments. We have updated Figure 2 to now also show the identified cortical and subcortical ROIs of the other participant. Furthermore, we have rephrased the specified line (now line 227-228) following the reviewer's suggestion in the revised manuscript.

2) Figure 4: Why are data (boxes) missing in a subset of the panels/ conditions? Could this be clarified in the legend and/or main text?

A missing box indicates that the given ROI could not be determined in the corresponding participant (i.e., no significant activation was observable in the whole-brain analyses at $q(\text{FDR}) < 0.05$). This explanation has been added to the revised legend of Figure 4 (p. 24).

3) Additional analyses to probe for specificity of the frontal alpha dependent BOLD responses in the motor system.

3.1.) The authors have tested a further set of ROIs to show that frontal alpha does predict to a lesser extent the BOLD responses in areas outside the motor system, which is important (I count 3 control areas x 2 sides = 6?). However, I would recommend to reconsider how this data is presented (a bit messy as is, lines 252 - 259). Could the results be summarized in one sentence simply stating the percentage of significant correlations/ comparisons (as done in the sentence above on null-events)?

We thank the reviewer for this useful suggestion and have now summarized the findings in an additional sentence, by listing the suggested percentages (in line with the other ROI results presented earlier in the manuscript; p. 8).

3.2.) It would seem important as well to test the specificity to frontal alpha by examining how a non-frontal EEG alpha signal would predict BOLD responses in motor areas. The parietal alpha signal (that has already been extracted, lines 546-547) could serve for this purpose.

The comparison between frontal and parietal alpha was included in the original manuscript: as expected, frontal and parietal alpha analyses generated highly comparable results.

However, following an earlier suggestion by reviewer 1, these analyses were subsequently removed from the revised manuscript. Reviewer 1 correctly indicated that the much larger magnitude of parietal alpha power can obscure the much weaker frontal (premotor) alpha. That is, truly frontal alpha activity is highly difficult to unambiguously distinguish from parietal alpha influences on frontal electrodes. Therefore, we focused on parietal alpha in the revision. In short, the suggested inclusion of the parietal alpha signal was already present in the revised (and original) manuscript. To avoid any potential confusions, we have replaced “alpha” by “parietal alpha” on several occasions throughout the revised manuscript.

4) Does Suppl Fig 4 reflect activity over FC4?

Suppl Fig 3 indeed reflects frontal beta power. The corresponding legend in the revised version now mentions this explicitly (Supplemental Material p. 3)

5) There are a few issues with sentence constructs/ language, e.g. line 156. Please check.

We have carefully checked our manuscript and rephrased various sentences (including previous line 156, now line 205-206) in the revised manuscript.